

# Boundary-layer turbulent processes and mesoscale variability represented by Numerical Weather Prediction models during the BLLAST campaign

Fleur Couvreux[1], Eric Bazile[1], Guylaine Canut[1], Yann Seity[1], Marie Lothon[2], Fabienne Lohou[2], Françoise Guichard[1], Eric Nilsson[2,3]

[1]CNRM (Météo-France and CNRS), Toulouse, 31057, France
[2]Laboratoire d'Aérologie, University of Toulouse, CNRS, Toulouse, France
[3]Uppsala University, Stockholm, Sweden

*Correspondence to*: Fleur Couvreux (fleur.couvreux@meteo.fr)

**Abstract.** This study evaluates the ability of three operational models, AROME, ARPEGE and ECMWF, to predict the boundary-layer turbulent processes and mesoscale variability observed during the Boundary Layer Late-Afternoon and Sunset Turbulence (BLLAST) field campaign. AROME is a 2.5km limited area non-hydrostatic model operated over France, ARPEGE a global model with a 10km grid-size over France and ECMWF a global model with a 16km grid-size. We analyze the representation of the vertical profiles of temperature and humidity and the time evolution of near surface atmospheric variables as well as the radiative and turbulent fluxes for a total of 12 24h-long Intensive Observing Periods. Special attention is paid to the evolution of the turbulent kinetic energy that was sampled by a combination of independent instruments. For the first time, this variable, which is a central variable in the turbulence scheme used in AROME and ARPEGE, is evaluated with observations.

In general, the 24h-forecasts succeed in reproducing the variability from one day to the other in term of cloud cover, temperature, boundary-layer depth. However, they exhibit some systematic biases, in particular a cold bias within the daytime boundary layer for all models. An overestimation of the sensible heat flux is noted for two points in ARPEGE, partly related to an inaccurate simplification of surface characteristics and over-predominance of forests. AROME shows a moist bias within the daytime boundary layer, consistently with overestimated latent heat fluxes. ECMWF presents a dry bias at 2 m above surface and also overestimates the sensible heat flux. The high-resolution model AROME better resolves the vertical structures, in particular the strong daytime inversion and the evening thin stable boundary layer. This model is also capable to capture the peculiar observed features, such as the orographically-driven subsidence and a well-defined maximum in water vapor mixing ratio in the upper part of the residual layer that arises during the evening due to mesoscale advection. The mesoscale variability is analyzed and the order of magnitude is also well reproduced in AROME. AROME provides a good simulation of the diurnal variability of the turbulent kinetic energy while ARPEGE shows a right order of magnitude.



# 1 Introduction

Limited area numerical weather prediction models are used routinely for operational weather forecasting across the world. Due to the increasing resolution, it becomes important to evaluate their capability in reproducing the low-troposphere vertical profiles of temperature and moisture as well as their surface turbulent and radiative fluxes as they are more and more used for numerous applications such as predictions of road black ices or agro-meteorology for instance. Here we present the performance of those models on the representation of the near-surface variables and boundary-layer turbulent kinetic energy which has been largely unexplored.

Evaluation and improvement of models is often a motivation to deploy instruments in field campaign. However, field campaign observations are not so often extensively used to evaluate the representation of surface and boundary-layer processes by operational models. Atlaskin and Vihma (2012) use observations from a field campaign to evaluate NWP models. They focus on the representation of very stable conditions in the northern Europe and show a systematic warm bias during the coldest nights characterized by very stable conditions. Many studies have used field campaign data to evaluate the behaviour of various non-operational limited-area models. Steeneveld et al (2008) used data from three particular days of the CASES-99 field campaign to evaluate the impact of the boundary-layer scheme and the radiative scheme on the performance of three different limited-area models. LeMone et al (2013) used CASES-97 observations to evaluate various diagnostics of the boundary-layer depth applied on simulations of a mesoscale model. In parallel, evaluation of models has been carried out over permanent observing sites such as ground-based remote sensing observations from the Swiss plateau (Collaud Coen et al, 2014), from the Atmospheric Radiation Measurement (ARM, Morcrette 2002 or Guichard et al 2003) or Cloud-Net sites (Illingworth et al, 2007). In particular, the CloudNet project allows a systematic evaluation of clouds in different operational forecast models. For instance, Bouniol et al (2010) show that models tend to overestimate the cloud occurrence at all levels.

The Boundary Layer Late Afternoon and Sunset Turbulence (BLLAST) field campaign was conducted from 14 June to 8 July 2011 at Lannemezan in southern France, in an area of complex and heterogeneous terrain. A wide range of instrument platforms including full-size aircraft, remotely piloted aircraft systems (RPAS), remote sensing instruments, radiosoundings, tethered balloons, surface flux stations, and various meteorological towers were deployed over different surface types (Lothon et al., 2014). During this campaign, twelve fair-weather days were extensively documented by Intensive Observation Periods (IOPs). Those days correspond mainly to high-pressure fair weather situations. In this study, we take advantage of the large dataset provided by this campaign to evaluate the vertical structure of the boundary layer and its diurnal evolution represented in NWP models. Here, we also focus on the mesoscale variability that can occur in the area and how this impacts the observations locally as well as how this is reproduced by the model. Indeed, Acevedo and Fitzjarrald (2001) showed with observations complemented by a Large-Eddy Simulation (LES) that the spatial variability peaks in the evening transition and that land use and orography play a crucial role in setting temperature anomaly patterns. They also found that, around sunset, horizontal advection plays a secondary role compared to vertical divergence.





In addition to a better understanding of the processes involved in the transition, several recent studies assessed the behaviour of single-column models to represent the entire diurnal cycle by comparison to LES. Single-column runs is often used as a more simplified configuration than a full 3D simulation in order to highlight some deficiencies in the physics parametrization of the model and to test new developments. By comparing 1D model to LES on a case based on observations

at Cardington, UK (Beare et al., 2006) that covers the transition from early afternoon to the next morning, Edwards et al. (2006) show that 1D model had difficulties to represent correctly turbulence diffusivity during the afternoon transition which impacts on the mean profiles. More recently, Svensson et al. (2011) compared LES and single column models on the entire diurnal cycle of a CASES-99 case and show a faster decrease of the temperature in the afternoon temperature compared to LES. However, such evaluation has not been carried out for operational NWP models and have not used observations of

turbulence in the entire boundary layer. For example, observations of the turbulent kinetic energy are quite rare relatively to mean meteorological profiles, and often punctual (field campaigns), therefore the boundary layer parametrization based on a prognostic equation of the turbulent kinetic energy, which has been shown to perform better than first-order scheme (Holt and Raman, 1988), has only been evaluated through comparison to LES (Cuxart et al, 2006 for instance). Here, we will carefully analyse the turbulent kinetic energy which is a key parameter of the turbulent scheme (Cuxart et al, 2000) used in

the two French models evaluated.

Our objectives are i/ to evaluate the skills of operational NWP models to predict the whole diurnal cycle of the boundary-layer temperature and moisture and in particular the afternoon transition, ii/ to assess the representation of the turbulent kinetic energy by models for which the boundary-layer parametrization is based on a prognostic evolution of the turbulent kinetic energy, iii/ to evaluate the evolution of surface thermodynamic parameters for different covers.

Observations and evaluated models are described in section 2 as well as the methodology used to carry out the comparison. Results are presented in section 3 focusing on the general representation of the entire diurnal cycle : we separately analyse the reproduction of the energy balance at the surface, the surface meteorological variables, the boundary-layer characteristics and we end the analysis with a specific focus on the behaviour of the models during the afternoon transition. Discussion and conclusion end the paper.

## 25 2 Methodology

### 2.1 Observations

The observations used in this study have been acquired during the BLLAST field campaign and have been described in details in Lothon et al. (2014). Here, they are briefly summarized. They consist of measurements made by remote sensing (Doppler lidar, aerosol lidar, UHF wind profiler) and in-situ (automatic meteorological stations, soundings,

remotely piloted aircraft systems, manned aircraft) instruments. They have not been used in the assimilation system and can therefore be used for evaluation purpose without ambiguity. Table 1 summarizes all the types of data and measurements used





in this study with details on the resolution of the raw data, the estimated parameters and their sampling. In the following, we used the observations from the 12 IOP days of the field campaign (Lothon et al., 2014).

In total, 7 different sites were instrumented by eddy covariance system and radiometers, documenting various types of covers (wheat, grass, forest, moor, corn and more heterogeneous sites).Forest and grassland are the two main land types of the area while moor and urban surface types are intermediate and corn, wheat and bare soil are in minority (Hartogensis, 2015). A common procedure to retrieve surface heat fluxes from the raw data acquired at 10Hz was applied to all surface turbulence station measurements and provided surface turbulent and radiative fluxes at a 30min resolution (De Coster and Pietersen, 2012). These observations are used to evaluate the radiative and turbulent fluxes as well as the meteorological parameters simulated by the models close to the surface. Their locations are indicated in Fig. 1b by small yellow dots. For these sites, the wind was measured at different altitudes above the ground and has been interpolated to 10m for comparison to the models using a logarithmic profile and the measure of the wind stress close to the surface.

To describe the vertical profile of the boundary layer we use the radiosoundings (MODEM) launched four times per day (0000, 0600, 1200 and 1800 UTC) on the north-easternmost site (main site in the following, indicated by large orange dots in Fig. 1b), hourly radiosoundings (Vaisala RS92 probes) of the lower troposphere (up to 3 to 4 km, Legain et al, 2013) launched from the southern most launching site (4 km apart from the main site) and the vertical profiles obtained from the remotely piloted aircraft system (RPAS) SUMO (Reuder et al., 2012) that flew around the main site and provided from 4 to 10 soundings of the lower troposphere during the afternoon of the IOPs. Those measurements provide vertical profiles of temperature, water vapour content and horizontal wind. Boundary-layer depths are derived from those profiles as detailed in section 2.3. Boundary-layer depths derived from UHF and aerosol lidar data are also used.

The combination of various measurements that provide estimates of the turbulent kinetic energy was a specificity of this field campaign. The Doppler lidar (Windcube, manufactured by Leosphere, Gibert et al, 2012), measurements from ground towers, aircraft measurements and the turbulence probe mounted on the tethered balloon (Canut et al, 2015) all provide estimates of the variance of horizontal and/or vertical wind at high sampling rates (every 4 s for the lidar and 0.1s for the turbulence probe) and therefore estimates of the turbulent kinetic energy (*tke*).

## 2.2 Numerical weather prediction models

In this study we evaluate the behaviour of three Numerical Weather Prediction (NWP) models:

- two NWP models of Météo-France: (i) a global model, ARPEGE (Courtier et Geleyn, 1988) with a stretched horizontal grid of about 10 km x 10 km over France with a 4Dvar assimilation system and (ii) a limited area non-hydrostatic model, AROME (Seity et al., 2011), with a grid of 2.5 km x 2.5 km and a 3Dvar data assimilation system;

- the operational ECMWF IFS model with a horizontal grid size around 16 km x 16 km (Simmons et al, 1989).

Table 2 presents the main characteristics (horizontal resolution, number of vertical levels, PBL scheme, initialization time and forecast run, initialization of the land-surface properties) for the three models. Table 4 presents the main physiographic





characteristics (altitude, albedo, vegetation fraction and roughness length) of the points extracted from those models for the different IOPs.

For this field campaign, the AROME model was run in near-real time over a smaller domain (about a quarter of France) using lateral boundary conditions and initial conditions from the operational AROME. This allows to provide specific outputs for the 16 grid-points surrounding the main site (Fig 1b).

All models employ a terrain following hybrid sigma-pressure vertical coordinate. The vertical grid however differs from one model to the other (Table 2): ARPEGE has 70 vertical levels with about 11 levels within the first km (first level at 16m), AROME has 60 vertical levels with about 15 levels within the first km (first level at 10m), and ECMWF has 91 vertical levels with about 11 levels within the first km (first level at 10m). The time step varies from 1 min for the AROME model to about 10 min for ARPEGE and ECMWF. The models also differ by their different parametrizations. For the boundary-layer turbulence, AROME uses an Eddy-diffusivity Mass flux concept with the local turbulence (small eddies) represented by a turbulent kinetic energy (*tke*) prognostic scheme (Cuxart et al., 2000) with a non-local length-scale (Bougeault and Lacarrere, 1989) and the boundary-layer thermals and shallow convection represented by a mass-flux scheme (Pergaud et al., 2009). ARPEGE uses the same *tke* prognostic scheme (Cuxart et al., 2000) and uses a mass-flux scheme to represent shallow convection (Bechtold et al., 2001). ECMWF uses a Eddy-diffusivity Mass flux based on two updraughts (Köhler et al., 2011) and a non-local K profile for the boundary layer while shallow convection is handled by a separate bulk mass-flux scheme (Tiedtke 1989). The surface scheme is ISBA in ARPEGE (Noilhan and Planton, 1989; Giard and Bazile, 2000), AROME uses the surface platform SURFEX (Martin et al., 2014) and ECMWF uses the HTESSEL model (Balsamo et al, 2009). All models have the same longwave radiation scheme, the RRTM parametrization (Mlawer et al, 1997) but differ on the shortwave component: ARPEGE uses the RRTM parametrization while AROME has the Morcrette et al. (2001) parametrization. The radiation scheme is called every hour for ARPEGE and every 15 min for AROME. Note that, at the time of the field campaign, in the operational version of ARPEGE the radiation scheme was called every three hours and this induced an abrupt unrealistic decrease of the incoming shortwave radiation during the afternoon transition (not shown) that has now been corrected in the operational model with a hourly call. Concerning the cloud scheme, ARPEGE uses a distribution of relative humidity based on Smith (1990), AROME a distribution of the deficit saturation based on Bougeault (1982) and ECMWF uses a prognostic scheme (Forbes et al, 2011). In ARPEGE, there are 12 various vegetation covers but a low or high vegetation criterion is affected to each point to rapidly distinguish the points in term of stomatical resistance and roughness length (Table 2) while in AROME each grid is associated to a certain fraction of various vegetation types (culture, land, town, mixtures of crop and woodland, Landes forest or broad-leaves forest).

## 2.3 Methodology of comparison

Due to the resolution of each model, real surface heterogeneities, topography and local circulation, a perfect match between observations and model outputs is not expected. In particular, as shown in Lothon et al. (2014), a large variability on the observed surface fluxes exists among the sites (Fig 1) even at scales smaller than 2.5 km, the size of a grid box of



AROME (see for example the differences between the moor and the corn sites, or the grass and the wheat sites) that are mainly due to surface cover. However, the variability among observations and the differences between model outputs and observations provide inferences on the main drawbacks of the models. In this section, we detail how the comparison is conducted focusing on the temporal and spatial resolution of the different variables obtained from models and observations.

The simulated grid-points (and associated columns) surrounding the locations of the measurement sites have been extracted as shown in Fig. 1: 3 neighbouring grid points have been extracted for ARPEGE, 16 neighbouring grid points for AROME (a box of 10x10km² including all sites) and 9 neighbouring grid points for ECMWF. For ECMWF we evaluate both the analysis available every 6 hours as well as the operational forecast with 3-hourly outputs for the surface characteristics from the run launched at 0000 UTC while for the two other models we show the forecast launched at 0000 UTC with hourly

outputs. The forecast length, analysed here, was selected to be 24h. The atmospheric variables correspond to instantaneous fields sampled every hour for AROME and ARPEGE and every 6 hours for ECMWF. The diagnostics T2m (temperature at 2m), rh2m (relative humidity at 2m) and ws10m (horizontal wind speed at 10m) are obtained using an interpolation following Geleyn (1988) based on the Monin-Obukhov theory between the surface and the first model level for ARPEGE and IFS or calculated using a prognostic surface boundary-layer scheme for AROME (Masson and Seity, 2009).

In the model, the boundary-layer depth is the first level where the *tke* gets below 0.01 m² s⁻². In observations, various diagnostics allow to derive the boundary-layer depth :

i/ the height of maximum air refractive index structure coefficient (Jacoby-Koaly et al., 2002) is obtained from UHF data; it usually is an estimate of the inversion height as this criterion detects the level of a humidity vertical gradient

ii/ the first level below the previous height where the *tke* dissipation rate gets greater than a threshold ($10^{-3}$ m²/s³ ) is also

derived from the UHF data; this criterion gives an estimate of the top of the turbulent layer,

iii/ the height of the largest gradient of aerosol backscatter from the aerosol lidar data (Boyouk et al, 2010); this is another way to estimate the inversion height and

iv/ the best (determined manually) of four criteria applied on the various vertical profiles from soundings and RPAS (Remotely Piloted Airplane Systems) (Lothon et al., 2014), using either the height where the virtual potential temperature

exceeds the averaged value over the lower levels plus 0.2, the height of maximum relative humidity, the height of maximum first derivative of the potential temperature or the height of minimum first derivative of specific humidity. Often, the criterion based on the virtual potential temperature is retained. The comparison of different boundary-layer depths derived from various instruments has been illustrated in Bennett et al (2010).

The decrease of the boundary-layer depth in the afternoon transition is a delicate process and in practice, its

estimation depends on the criteria used to derive the boundary-layer depth as already shown by Angevine and Grimsdell (2002). This will be detailed in Sect. 3.5. The diagnostic used in the model has been compared to the criteria iv applied on the model profiles. In ARPEGE, the model diagnostic tends to overestimate the value derived from the profiles of about 200m while in AROME there is a very good agreement except for 14 June after 1500UTC, 15 June after 1400 UTC and 26





June due to the presence of clouds (discussed later). Therefore in the following, we will use the model diagnostic discarding those hours of disagreement.

When comparing observations and modelling, we have taken into consideration the fact that the horizontal and temporal average in observations should be as consistent as possible with the time step and resolution of simulations. In the latter, the surface turbulent and radiative fluxes at hour h correspond to the average value between hour h-1 and hour h. In the observations, values have been processed every 30min and are then averaged to provide 1hr-average for the comparison. Furthermore, one must keep in mind the reduced surface (few hundred metres footprint) sampled in the measured surface turbulent fluxes compared to the grid size of the three NWPs.

Concerning the *tke,* in the observations, it has been estimated for 20 min time windows for the 60m-tower, the Doppler lidar and the tethered balloon, 10 min for the 10m-tower (sensitivity to a computation with 20min did not change the results) and for horizontal legs of 25-30 km for the aircraft measurements (corresponding to 5 min cf Table 1 and Canut et al, 2015 for more details; this is a compromise between having the same time window as the other measurements and minimizing the influence of the mesoscale heterogeneities). Note that a 5km high-pass filter has been applied only to the aircraft raw data before the calculation of the *tke* to filter out the mesoscale variability. We also tested the *tke* estimates obtained with 2.5km high-pass filter but it was affected by a large time-variability which highlighted that the samples were not large enough.

In the models, a horizontal resolution of 2.5km and 10km respectively in AROME and ARPEGE is equivalent to 9 and 30 min respectively according to a wind speed around 3-5ms$^{-1}$ in the boundary layer, which is consistent with the 20 min used to derive the *tke* from surface point observations. The estimation of the *tke* with the Doppler lidar (Gilbert et al, 2012) assumes that the turbulence is isotropic and derives the value from the measured vertical velocity variances. To evaluate this hypothesis, we compute the ratio

$$A = 1.5 \frac{\overline{w'^2}}{tke}$$

a coefficient from the tower measurements (both from the 60m tower and the 10m tower) and from the tethered balloon, A=1 if the turbulence is isotropic. When A>1, the contribution of the vertical velocity variance is dominant (A=3 if the horizontal velocity variances are null). When A<1, the contribution of horizontal variance is dominant. Both the tower measurements as well as the tethered balloon[1] measurements indicate that above 0.1 to 0.2 zi, zi being the boundary-layer height, and in the middle of the boundary layer, this coefficient is between 1 and 2 suggesting that the variance of the vertical velocity is often the main contributor to the *tke* at that height and the *tke* can be estimated from the $\overline{w'^2}$ as $tke = 1.5 \overline{w'^2}$ . A sensitivity to this ratio for the estimation of the *tke* is indicated in the Appendix. Aircraft measurements indicate that closer to the top of the boundary layer this coefficient decreases again with value between 0.75 and 1. Below 0.1 zi, the variance of horizontal wind is important and this coefficient is mostly below 0.6 (see Canut et al, 2015 for more details). Therefore, in

---

[1]The tethered balloon never reaches height above 500m





the following, we only use Doppler lidar estimates from altitudes above 100m. More complex computations taking into account the day-to-day and vertical variation of the anisotropy factor derived from tethered balloon or aircraft could be done in a future study. Note also that as we derive the *tke* as 1.5 $\overline{w'^2}$ we tend to overestimate the observed *tke* most of the time but we may underestimate it on days with more wind, conditions in which horizontal wind fluctuations are expected to be larger.

Eventually, in order to characterize the afternoon transition (AT), the time at which the sensible heat flux gets negative is determined in both observations and models. This is done by finding the 0-cross over from the interpolation of hourly flux outputs.

In the following, we evaluate the representation of the diurnal cycle of the boundary-layer characteristics and surface energy budgets over all IOPs.

## 3 Results

In this section, we compare surface fluxes, meteorological variables, boundary-layer structure, turbulent kinetic energy for the 12 IOP days. As shown in Lothon et al (2014), those days correspond to mainly high-pressure fair-weather conditions with no cloud cover or a small amount for 14, 15, 24 and 30 June. Most of the days experienced a typical mountain breeze circulation with nocturnal southerly down-slope wind and north-westerly to north-easterly up-slope wind during the days. The 25, 26 and 27 June did not register such circulation (cf Lothon et al, 2014, Fig 6) and were characterized by easterly winds. These three days also showed higher temperature and stronger wind which was due to the presence of a low pressure in the Gulf of Lion (for more details see Nilsson et al, 2015a). In the following, those three days will be referred to as hot days.

### 3.1 Radiative and surface fluxes

Figure 2 presents series of 24h sequences for the 12 different IOPs (from 14 June to 5 July 2011), of the observed and simulated surface downwelling solar radiation. In Figure 2a, the different model grid points are plotted as well as the dark grey shading that indicates the envelope containing the different surface sites, which quantifies the spatial variability. Figure 2b shows the mean value and the maximum range[2] for a given type (observations or models) averaged for daytime and nighttime respectively as a measure of the spatial variability. The cloudy days are clearly depicted by an increase in the spatial variability of the observed surface downwelling solar radiation (Fig 2a) consistently with Lothon et al (2014). ARPEGE and AROME mostly distinguish between the clear-free days (noted 'C') and the cloudy days indicated by an increase spatial variability (Fig 2b). ECMWF for at least two observed clear days (20 June, 27 June) depicts a decrease of downwelling solar radiation from 1030 to 1330 UTC which suggests the presence of clouds in the model. The 26 June has

[2]This is computed at each time step by the difference between the maximum and the minimum over all the points of the given type



some clouds from 1400 UTC to 1900 UTC while ECMWF predicts variability in the downwelling solar radiation from 1030 to 1330 UTC. The 27 June has high clouds in ARPEGE throughout the day while observations only registered thin cirrus after 1700 UTC (not shown). The 30 June presents stratocumulus in the morning that clear up through the afternoon with however quite a variable cloud cover in the afternoon while ARPEGE and ECMWF predict a cloud-free atmosphere. The spatial variability is slightly overestimated for 14, 15, 30 June in AROME but otherwise in good agreement with observations. In summary, all models capture in general the spatial and temporal variability in downwelling solar radiation with however a better behaviour for AROME in terms of cloud occurrence and spatial variability.

There is more discrepancy in the simulations of sensible heat fluxes with biases reaching more than 100 Wm$^{-2}$ (Fig 3a). First, ARPEGE predicts very large sensible heat fluxes which have similar range as observations above the forest (dashed and dash-dotted black lines in Fig 3a) for two of the three points (ARP1 and ARP3 in Table2 which mainly differ from ARP2 in terms of altitudes and roughness lengths) : those two model grid-points are characterised by high vegetation cover which have lower albedo (0.12 against 0.2); they are also at higher altitude. These simulated sensible heat fluxes are too large values to be representative of a 10km wide grid box over the area which is characterized by much more surface heterogeneities at this size (cf Fig 1). The third point (northernmost, ARP2) is in better agreement with the non-forest sites (indicated by the grey shading). ECMWF overestimates the surface sensible heat fluxes. The variability from one IOP to the other (Fig 3b) is correctly reproduced by all three models with, for instance, a decrease of the maximum sensible heat flux during the hot days. They also all predict more negative sensible heat flux during the nights of the hot period (from 25 to 27 June) even though ECMWF and ARPEGE underestimate this negative sensible heat flux while AROME overestimate the value in the first night (25 to 26 June). Concerning the spatial variability, one can note the large value obtained from the surface sites. The observed range is computed either for all the stations (full black line) or by removing the forest stations (dash-dotted black line). The forest stations induce larger observed range especially during the first part of the period. The spatial variability among the various ECMWF grid-points is much smaller which is partly explained by a coarser horizontal grid-size while the value for ARPEGE and AROME is of the same order of magnitude as the observations but slightly underestimated at the end of the period.

Latent heat fluxes predicted by AROME systematically overestimate the observed values by up to 100 Wm$^{-2}$ (Fig 3c) and this may be related to a too large soil moisture content (however, no observations were available at various sites to evaluate this variable). The two high-vegetation points of ARPEGE do not tend to evaporate more as could have been expected from a larger net radiation (due to a lower albedo). ECMWF correctly reproduce the range of observations. The variability among the various IOPs is also correctly reproduce with higher latent heat fluxes during the hot days (Fig 3d). The spatial variability is about the same order of the observed one in AROME, slightly underestimated in ARPEGE and strongly underestimated in ECMWF. Interestingly, when plotting the latent heat fluxes as a function of the sensible heat fluxes at 1200 UTC, the models reproduce the -1 slope related to an almost constant available energy (cf Supplementary Fig 1) in agreement with LeMone et al (2003). This is more valid for the clear days (cyan or blue symbols) versus the cloudy days (green and purple symbols) in agreement with Lohou et al (2014). Most of the observations also record a negative



relationship (even though with a less steep slope) except the observations at 60m on the tower (grey squares) and observations at 30m over the forest (dots).

In summary, one can note an overestimation of the sensible heat flux by ARPEGE for the two points with high vegetation and by ECMWF in a lesser extent and an overestimation of the latent heat flux by AROME (strong bias). All models underestimate the observed spatial variability. This underestimation is larger for ECMWF probably due to the larger horizontal grid-size and more expanded area for the 9 extracted grid-points.

## 3.2 Meteorological variables

Figure 4 presents the same figures as Figure 3 for the 2m temperature, 2m water vapour mixing ratio and the 10m wind speed observed and simulated. AROME and ARPEGE are in very good agreement with the observed close to surface meteorological variables. First, all models reproduce the variability, through the period, of the 2m temperature with in particular a warming period from 24/06 to 27/06. In AROME and ARPEGE, the maximum of daytime temperature occurs earlier (by about one hour) than in the observations (note that this can not be analysed in ECMWF with a 3-hourly outputs). The main discrepancies occur during the night where the models tend to have a cold bias consistently with common deficiencies of NWP models (Svensson et al, 2011). Interestingly, the spatial variability in night time temperature among sites is smaller for the hot period; this might be due to larger wind speed during this period. The models do not reproduce this behaviour: during the hot period, the model predicts both an increasing variability of night sensible heat fluxes and 2m temperature. The underestimation of the spatial variability by AROME and ARPEGE during most days is not due to a misrepresentation of the wind as the wind is relatively weak over the whole period and in more or less agreement with observations. ECMWF overestimate the spatial variability which is partly explained by the westerly grid points being warmer (not shown). Also the diurnal cycle of the spatial variability in ECMWF is inverse compared to the observations with higher daily variability than nightly variability. This needs further investigation.

Concerning the 2m water vapour mixing ratio, the models reproduce the increase that follows a precipitating events (indicated by the double vertical dotted lines). The models also reproduce the increase in spatial variability during the hot period. There is no clear diurnal cycle except in ECMWF which overestimates the range of variations from night to day and the spatial variability. Also ECMWF has a dry bias during daytime especially in the second part of the period. One can note that the overestimation of the latent heat fluxes by AROME has no clear consequences in the reproduction of the 2m water vapour mixing ratio. Concerning the 10m wind speed ARPEGE & AROME reproduce larger wind speed (greater than 2-3 ms$^{-1}$) during the hot period with also a larger spatial variability. ECMWF does not reproduce this shift.

In summary, one can note a very good simulation of the surface meteorological variables in AROME and ARPEGE; it is slightly less accurate in ECMWF especially for wind speed and water vapour mixing ratio. In the following sections, we only focus on the French models for which we have hourly outputs.





### 3.3 vertical structure

Thanks to the numerous soundings of the atmosphere via various techniques (radiosoundings, low-atmosphere radiosoundings or RPAS profiling), it is possible to extensively evaluate the evolution of the boundary-layer vertical structure predicted by the models.

Figure 5 presents scatterplots of the simulated versus observed values of the potential temperature and water vapour mixing ratio averaged over the first 500m deep layer. First, there is a good agreement between all types of observations for potential temperature. Then, the MODEM soundings are drier than the others about 1 g kg$^{-1}$ consistently with Agusti-Panareda et al (2009). AROME and ARPEGE display a cold bias of about 1.5K. In ARPEGE, the temperature bias is dependent on the average temperature with no more bias for temperature greater than 305K. ARPEGE does not present a

warm bias despite its overestimation of the sensible heat flux for two of the grid-points. AROME presents a moist bias which is consistent with the too high latent heat flux while ARPEGE exhibits a dry bias. The AROME moist and cold biases were not clear in the time evolution of 2m variables indicated different reproduction of the surface layer versus the boundary layer.

Figure 6 illustrates the time evolution of the vertical profiles of potential temperature and water vapour mixing ratio (sampled every two hours for clarity) from 12 to 20 UTC for two clear IOP days the 27 June 2011 (one of the hot days) and 

the 1 July 2011. AROME captures better the strong inversion in potential temperature that occurs at the top of the boundary layer (at 1400 the 27 June or the 01 July) and this is true for most of the IOPs. This may be due to the finer vertical grid. In both models, there are more spatial variability during the hot period than otherwise and this remains true throughout the entire day, this is consistent with the results at the surface (higher variability in terms of surface heat fluxes and 2m-meteorological variables) as shown previously. In particular in AROME, the 27 June, the variability among the 16 columns

is larger than the variability among the 3 ARPEGE columns even though the area covered by the 16 AROME points is equivalent to one grid size of ARPEGE. One can note for the 01 July the maximum in water vapour mixing ratio in the upper part of the boundary layer simulated by AROME which is also observed in the radiosoundings. Analysis of the moisture budget indicates that this maximum is mainly related to advection (not shown) suggesting that mesoscale circulation has an impact on this peculiar boundary-layer structure.

To further assess the representation of the vertical structure of the boundary layer, we compare the boundary-layer depths estimated by the model with boundary-layer depths estimated with observations. Figure 7 presents the time evolution of the different boundary-layer depth estimates for all the IOPs. The overestimation of the boundary-layer depth for AROME and ARPEGE (more pronounced in ARPEGE) on 14 and 15 June 2011 is explained by the modelled boundary-layer depth criterion based on significant *tke* that depicts the top of the shallow cumulus layer. Both AROME and ARPEGE are able to

reproduce the temporal variability in terms of maximum boundary-layer depth from one day to the other with for instance a shallower boundary layer during the hot days and, the highest on 30 June, 1 July and 2 July if we discard the 14 and 15 June. The model forecasts are initialized every day so part of the variability among the IOPS is forced through the initial state, but the existence of variability of the boundary-layer depth among the IOPs highlights that the physics of the models respond





correctly to these differences in weather. Lothon et al (2014) identified three types of growth of the boundary layer occurring in the morning of the day: typical growth the 20, 24, 25, 30 June and 02 July, slow growth the 26 June, 27 June and 05 July and rapid growth the 14, 19 June and 01 July. This distinction is reproduced by the models. Evaluating the decrease of the boundary layer in the afternoon is more complex. The aerosol diagnostic based on the lidar measurement always depicts the

top of the inversion layer in the afternoon while the profile diagnostics as well as the reflectivity gradient from the UHF indicate either the top of the stable layer or the top of the inversion layer depending on the cases. The model diagnostic depicts the top of the turbulent layer which is also the case of the boundary-layer depth diagnosed from the dissipation rate measured by the UHF. The difference between those diagnostics in the afternoon depicts the existence of a pre-residual layer in-between the top of the turbulent layer and the top of the inversion layer as detailed in Nilsson et al (2015b). Concerning

the decrease, ARPEGE most of the time predicts a later decrease than AROME. AROME is in better agreement with the boundary-layer depth diagnosed from the dissipation rate even though AROME tends to be slightly higher, which could also be explained by the fact that the turbulence variable used to diagnose the boundary-layer depth is different: tke versus dissipation. Eventually, one can note the large spatial variability among the model grid-points in particular 26 and 27 June and 02 and 05 July, however the highest boundary layer is not systematically always over the same grid point, so this can not

be explained by particular surface characteristics.

### 3.4 turbulent kinetic energy

A specificity of this campaign is the existence of various simultaneous measurements of the turbulent kinetic energy at various heights in the atmosphere. We use those measurements to evaluate the reproduction of the *tke* by the subgrid turbulence scheme in AROME and ARPEGE.

Figure 8 presents the time evolution of the *tke* for all the IOP days close to the surface and higher in the boundary layer. In the upper panel, the *tke* observed close to the surface, at ~ 8m, is compared to the modelled *tke* at the first level (at 11m in AROME and 17.5m in ARPEGE). Often, observations show significant *tke* in the morning that is not simulated except for a few days (25, 26 and 27 June for AROME and 24 June for ARPEGE) characterized by a larger wind speed and therefore a stronger shear production (Fig4e). There is also significant *tke* in the evening with a minimum around sunset that

is also not simulated except for a few days (20, 25, 26 June and 5 July for AROME and 5 July for ARPEGE). This minimum of *tke* is associated to a minimum of wind speed which is present for most days with weak wind. Note that the maximum measured the evening of the 27 June is associated to convective storms and are reproduced by the models. Those morning and evening *tke* values are related to slope-wind and also potentially effect of nocturnal low-level jet in the early morning. ARPEGE tends to present a Gaussian diurnal cycle of the *tke* most of the days (except 3 days : 24 June, 27 June and 05 July

where maximum of *tke* exist in the morning or the evening) but with a maximum value consistent with observations. AROME systematically underestimates the maximum value but records a variable diurnal cycle from one day to the other. This underestimation is in apparent contradiction with a larger sensible heat at least in the end of the period. The higher value in ARPEGE can be explained by a higher model level (17.5m versus 11m, as less turbulence is expected close to the





ground), a larger grid size (9km versus 2.5km). Higher in the atmosphere, the modelled *tke* and observed one are in better agreement. Note that the various types of observations agree together in terms of intensity. The temporal variability at those levels is well reproduced by the models with smaller values during the hot period in agreement with lower buoyancy flux which is the main source of the *tke* during the day (see also Nilsson et al, 2015). At 60m and higher up, AROME has

systematically less *tke* than ARPEGE probably for the same reasons as for the low levels.

Figure 9 illustrates the time evolution of vertical profiles of the turbulent kinetic energy modelled and observed for the 1 July (this is the only day where we have enough observations to retrieve a time-varying vertical profile of the *tke*). AROME has lower *tke* than ARPEGE and it decreases the turbulence earlier (starting at 1400) than ARPEGE (starting at 1500) as also shown in Fig 8. The shape of the vertical profiles is consistent among each model and the observations. The

lidar observations (triangles, note that this a *tke* estimate deduced from the turbulent variance of the vertical velocity) indicate a more or less stationary value in the middle of the boundary layer from 1400 to 1600 which is not simulated by the models. However, reminds that the lidar only measures the vertical velocity variances and therefore neglects any fluctuations in the horizontal velocity variances. But the comparison of the squared (tethered balloon) and the triangle (Doppler lidar)symbols of the same colour and at the same altitude provides an estimation of the error obtained from this estimation :

A is underestimated during daytime with values more around 1.3-1.8 while A is overestimated in late afternoon (1700 and 1800) with A around 0.4-0.8. This deserves further investigation with more measurements of the vertical profiles. Also the contribution of the shear contribution versus the buoyancy contribution in the creation of *tke* could be further analysed in observations and models and in general the budget of *tke*.

**3.5 Afternoon transition**

In this section, we focus on the afternoon transition period. Most of the physical processes, including turbulent ones, are small and on the same order of magnitude during the later part of the transition and the turbulence regime changes from the fully convective regime of turbulence, close to homogeneous and isotropic, towards more heterogeneous and intermittent turbulence.

Concerning the evolution of the boundary layer in the afternoon, the IOP days can be separated in the two

categories proposed by Grimsdell and Angevine (2002) as depicted by the behaviour of the UHF reflectivity with 24/06, 30/06, 01/07 and 02/07 pertaining to the inversion layer separation cases (ILS, so-called by Grimsdell and Angevine, 2002, where the reflectivity gradient stays more or less at the same height as the maximum registered during the day) and 25/06, 26/06, 27/06 pertaining to the descent cases (where the reflectivity gradient decreases with height in the evening). As in Grimsdell and Angevine (2002), the ILS cases are colder and drier days characterized by strong inversion at the top of the

boundary layer in potential temperature and associated strong shear as shown in Nilsson et al (2015a) ; those cases have also a strong inversion reproduced by models (not shown). The descent cases are warmer and moister days corresponding to the hot period. However, the height of the strongest gradient in the UHF reflectivity is more representative of the top of the inversion layer and does not really determine the top of the turbulent layer which is better depicted by the height derived





from the dissipation rate (in pink in Fig 6). This latter height is more comparable to the boundary-layer depth diagnosed in the models. AROME always predicts an earlier decrease of turbulence than ARPEGE and better agrees with the evolution of the height derived from the dissipation rate. The layer between the pink and the red symbols has been denoted as a pre-residual layer by Nilsson et al, (2015b). It is characterized by very low turbulence and results from the adjustment of turbulence to the decreasing surface fluxes (Darbieu et al, 2015).

Figure 10 presents the variations of the time when the virtual temperature flux (which is a combination of the surface sensible heat flux and the latent heat flux) goes negative, t_Hv0, through the IOPs and the various points. This time strongly varies in the observations from one surface to the other as already shown by Lothon et al (2014, their Fig 8 and black symbols in Fig 9), suggesting that the vegetation partly drives the delay of the transition from one site to the other. The range of t_Hv0 among the three points of ARPEGE (blue symbols) is less than one hour except during the hot period (26 and 27 June) and 5 July. The range of t_Hv0 is much larger in AROME (cyan symbols) with a range varying from 2 hours to 6 hours with however no systematic behaviour for a given point (indicated by a given symbol). AROME systematically has an earlier t_H0 than ARPEGE consistently with an earlier decrease of turbulence. Also this time occurs earlier during the hot period and this is reproduced by the models. In observations and models, the spatial variability is the strongest during the hot period.

In summary, the models are doing a relatively good job during the afternoon. This could be related to the quasi-stationnary behaviour discussed in Darbieu et al (2015) and Nilsson et al (2015) where no change in turbulence structure or characteristics once normalised by the decreasing surface sensible heat fluxes. The difficulties are picking up in the very late afternoon. We have also indicated more difficulties in the models to reproduce the varying characteristics of close to surface variables at night highlighting difficulties in the models to reproduce correctly the stable conditions.

## 4. Conclusions

The BLLAST field campaign gathered a large dataset, in particular high-frequency observations of the vertical structure of the boundary layer and observations of the turbulent kinetic energy that enables us to extensively evaluate three numerical weather prediction models. In summary, all models reproduce the temporal variability observed among the different IOPs in terms of variations of the cloud amount (clear versus cloudy conditions), of maximum height of the boundary layer, of variations of temperature. This is also a necessary first step if we want to further use those models to derive large-scale fields such as large-scale advection that are needed for smaller scale modelling studies. For instance, during the hot period, models and observations predict less sensible heat fluxes, larger temperature, larger wind speed, less *tke*. The different types of growth of the boundary layer encountered during the field campaign and detailed in Lothon et al (2014) are correctly distinguished by AROME and ARPEGE. However, systematic biases appear over those 12 IOPs: too large latent heat fluxes in AROME, a too large relative humidity diurnal amplitude at 2m and a dry bias during the day for ECMWF (especially at the end of the period). For two ARPEGE points the surface fluxes are similar to measurements over





forest whereas the satellite map does not indicate a homogeneous forest patch over 10x10km² in the area. AROME better reproduces the vertical structures as well as the variability among the different IOPs in boundary-layer depth in terms of daily maximum value or growth in the morning. The spatial variability reproduced by AROME is similar from the one derived from the various in-situ surface sites.

For the first time, the turbulent kinetic energy, the prognostic variable of the turbulence scheme in AROME and ARPEGE, has been evaluated. Both models reproduce the right order of magnitude. AROME better reproduces the variation from one day to the other of its diurnal cycle while ARPEGE always predicts a similar bell shape evolution. However, AROME underestimates the value while ARPEGE is in better agreement with the observed intensity. This may be due to difference in grid-size but also in physical parametrization. The EDMF scheme used in AROME predicts the contribution of

the thermals to the turbulence by a mass-flux scheme which indirectly feedbacks the *tke* via the thermal production term (the mass-flux scheme contributes to the buoyancy flux profile) whereas in ARPEGE, the turbulence is only reproduced by a *tke* prognostic scheme. In a future study, we could gain some insight by evaluating the different simulated terms of the near-surface *tke* budget that has also been derived in observations in Nilsson et al (2015a).

       In summary, this study is a first attempt to analyse the improvements provided by high-resolution numerical

weather prediction. As such, AROME seems to better depict the mesoscale spatial and temporal variability. However, future studies are needed in order to determine the exact role of the increase in resolution versus the change in physical parametrization.

**Appendix**

In order to evaluate the sensitivity of the determination of the tke using the $<w'^2>$ variance, we plotted in Supplementary

Figure 2, the time evolution of the tke at two different vertical levels (200 m and 450 m) derived from Doppler lidar measurements using three different values of A. However comparison of this figure and Figure 9 indicates that A=3 is a much too large values.

**Acknowledgements**

       The authors thank F. Said for providing the tower measurements, J. Reuder for providing the SUMO measurements,

F. Gibert for providing the lidar measurements, P. Augustin for the boundary-layer diagnostic derived from the lidar, E. Pardyjak, D. Alexander and C. Darbieu for the forest flux measurements, D. Legain for the contribution of the CNRM to the field campaign, P. Durand for the Piper Aztec turbulence data process, B Piguet for the tethered-balloon turbulence data process, LEOSPHERE for providing the Doppler lidar to the field campaign. The BLLAST field experiment was made possible thanks to the contribution of several institutions and supports: INSU-CNRS (Institut National des Sciences de

l'Univers, Centre National de la Recherche Scientifique, LEFE-IMAGO program), Météo-France, Observatoire Midi-



Pyrénées (University of Toulouse), EUFAR (European Facility for Airborne Research), BLLATE-1&2, COST ES0802 (European Cooperation in the field of Scientific and Technical). The field experiment would not have occurred without the contribution of all participating European and American research groups, which all have contributed in a significant amount. The Piper Aztec research airplane is operated by SAFIRE, which a unit supported by INSU-CNRS, Météo-France and the

French Spatial Agency (CNES). BLLAST field experiment was hosted by the instrumented site of Centre de Recherches Atmosphériques, Lannemezan, France (Observatoire Midi-Pyrénées, Laboratoire d'Aérologie).

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

5   **Tables:**





Table 1. List of the instruments and their spatial and temporal resolutions

| Instrument | Used measured parameters | Derived diagnostics | Time resolution/range | Spatial resolution/range | Location |
|---|---|---|---|---|---|
| Standard radiosoundings (MODEM) | q, $q_v$, wind speed | $h_{BL}$ | 0000, 0600, 1200, 1800 UTC | ~10-15m/0-20km | Main site |
| Low-troposphere soundings | q, $q_v$, wind speed | $h_{BL}$ | Hourly from 1200 to 2200 UTC in IOP | ~10-15m/0-2km | |
| Turbulence station (eddy-covariance system) | T2m, q2m, ws10m, sensible & latent heat flux, $u'^2$, $v'^2$, $w'^2$ | | 30 min from 20 Hz (except the forest site that has 10 Hz) sampling rates | | 7 stations over wheat, grass, forest, moor, corn |
| Radiative flux station (radiometers) | incoming & outgoing shortwave and longwave radiation | | 1 Hz sampling rates | | Moor, Corn, Forest, main tower sites |
| UHF | refractive index structure coefficient, Turbulent energy dissipation rate | $h_{BL}$ | 5 min consensus (2 cycles over 5 beams) | ~75m /175m-4000m | |
| Doppler lidar | Vertical velocity | tke | 4s time resolution; turbulence moments calculated on *20* min | 50m | |
| Aerosol lidar | Aerosol backscatter | $h_{BL}$ | 4s time resolution but diagnostic derived every 15 min | 15m | Main site |
| French Piper Aztec aircraft | 3-D wind | tke | 25 Hz high rate measurements moments calculated on 5-7 min samples | ~3m spatial resolution of the high rate measurements; aircraft velocity of 70 m/s; turbulence moments calculated over 30-40 km legs stabilized in attitude & altitude | |
| Remote piloted aircraft system SUMO | q, $q_v$, wind speed | | 2Hz for thermo and 100 Hz for wind | | Main site |
| Tethered Balloon with a turbulence probe | $u'^2$, $v'^2$, $w'^2$ | tke | 20 min from 10 Hz sampling rates | | Main site |



Table 2. Description of the three models

| Model | Horizontal resolution | Number of vertical levels in total and in the first atmospheric kilometer, first level altitude | time step (mn) | Surface scheme | PBL scheme | Initialization time/ model run length (hours) | Initialisation of land-surface properties |
|---|---|---|---|---|---|---|---|
| AROME | 2.5 km | 60 / 15 / 10m | 1 | SURFEX | TKE prognostic scheme – Mass flux scheme for dry and cloudy thermals | 00TU; 30 | From a surface reanalysis with this model |
| ARPEGE | 10 km | 70 / 11 / 16m | 10 | ISBA | TKE prognostic scheme – mass-flux scheme for cumulus | 00 TU; 36 | From a surface reanalysis with this model |
| ECMWF | 16 km | 91 / 11/ 10m | 10 | HTESSEL | Non-local K profile ; mass-flux for cumulus | 00-06-12-18 TU; 06 | From a surface reanalysis with this model |



Table 3. Surface characteristics of the various points extracted from the models

| Points | Altitude (m) | Albedo | Vegetation fraction | LAI | Roughness length |
|---|---|---|---|---|---|
| ARO-1 | 535 | 0.18 | 0.95 | 3.4 | 0.78 |
| ARO-2 | 611 | 0.19 | 0.93 | 3.5 | 0.53 |
| ARO-3 | 595 | 0.19 | 0.92 | 3.2 | 0.26 |
| ARO-4 | 558 | 0.20 | 0.92 | 3.4 | 0.16 |
| ARO-5 | 552 | 0.20 | 0.92 | 3.5 | 0.24 |
| ARO-6 | 605 | 0.19 | 0.93 | 3.4 | 0.38 |
| ARO-7 | 609 | 0.16 | 0.85 | 3.3 | 0.45 |
| ARO-8 | 593 | 0.17 | 0.94 | 3.2 | 0.39 |
| ARO-9 | 532 | 0.19 | 0.93 | 3.5 | 0.49 |
| ARO-10 | 567 | 0.19 | 0.91 | 3.7 | 0.37 |
| ARO-11 | 579 | 0.20 | 0.91 | 3.3 | 0.18 |
| ARO-12 | 575 | 0.19 | 0.91 | 3.5 | 0.47 |
| ARO-13 | 505 | 0.18 | 0.93 | 3.8 | 0.83 |
| ARO-14 | 521 | 0.18 | 0.92 | 3.7 | 0.64 |
| ARO-15 | 529 | 0.19 | 0.88 | 3.2 | 0.23 |
| ARO-16 | 527 | 0.19 | 0.90 | 3.5 | 0.38 |
| ARP-1 | 701 | 0.12 | 0.86 | 3.7 | 1.8 |
| ARP-2 | 477 | 0.2 | 0.84 | 3.2 | 0.17 |
| ARP-3 | 778 | 0.12 | 0.85 | 3.6 | 1.93 |
| ECMWF-1 | 1068 | 0.15 | Not available | Not available | 6.2 |
| ECMWF-2 | 894 | 0.15 | Not available | Not available | 5.1 |
| ECMWF-3 | 772 | 0.15 | Not available | Not available | 4.8 |
| ECMWF-4 | 510 | 0.15 | Not available | Not available | 0.65 |
| ECMWF-5 | 491 | 0.15 | Not available | Not available | 0.62 |
| ECMWF-6 | 463 | 0.15 | Not available | Not available | 0.88 |
| ECMWF-7 | 282 | 0.15 | Not available | Not available | 0.65 |
| ECMWF-8 | 314 | 0.15 | Not available | Not available | 0.62 |
| ECMWF-9 | 325 | 0.15 | Not available | Not available | 0.62 |



**Figures**




January, 2016

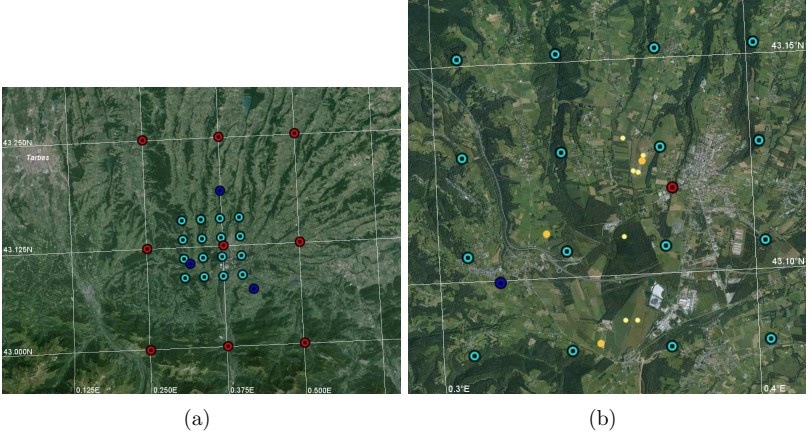

(a)                          (b)

Figure 1: (a) Map of the different points extracted from the models (red for ECMWF, blue for ARPEGE and cyan for AROME).(b) zoom of (a) with surface sites shown by small yellow dots and radiosoundings launching site in large orange dots. Note that the western most site was the site for launching the few GRAW soundings that were not used in this study (Google Earth Source).





January, 2016

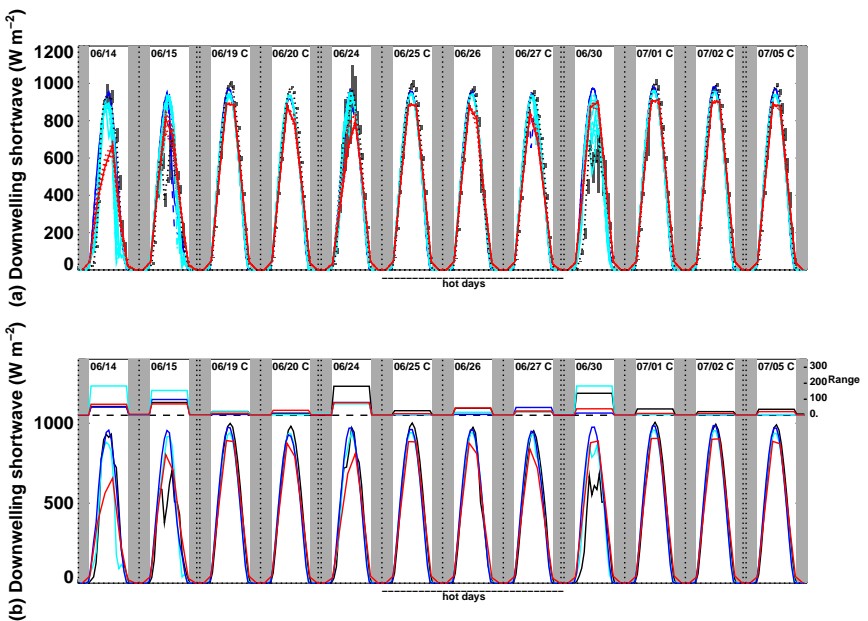

Figure 2: Time series of 24h sequences for the 12 IOPs of surface downwelling solar flux measured over several surfaces in black for the mean and the range (as attested by +/- the standard deviation) shaded in grey in (a), simulated by ARPEGE in dark blue, by AROME in light blue and ECMWF in red. The vertical grey shading marks the nighttime. (b) presents the mean value (left axis) and the maximum spatial variability (right axis), computed as the difference between the maximum value and the minimum value for all sites or all grid points of a given model but averaged respectively over day and night. Note that for ARPEGE, due to the different behaviour of ARP1 and ARP3, only ARP2 is plotted as a mean while the spatial variability is computed with the three points.




January, 2016

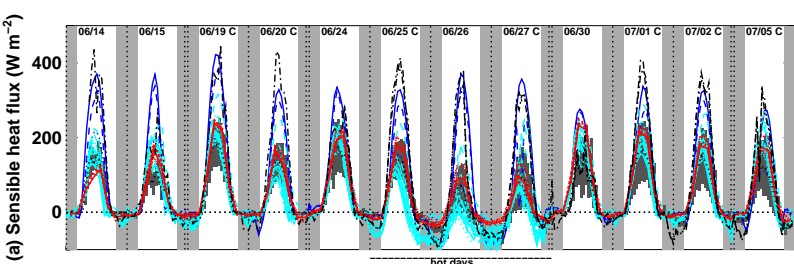

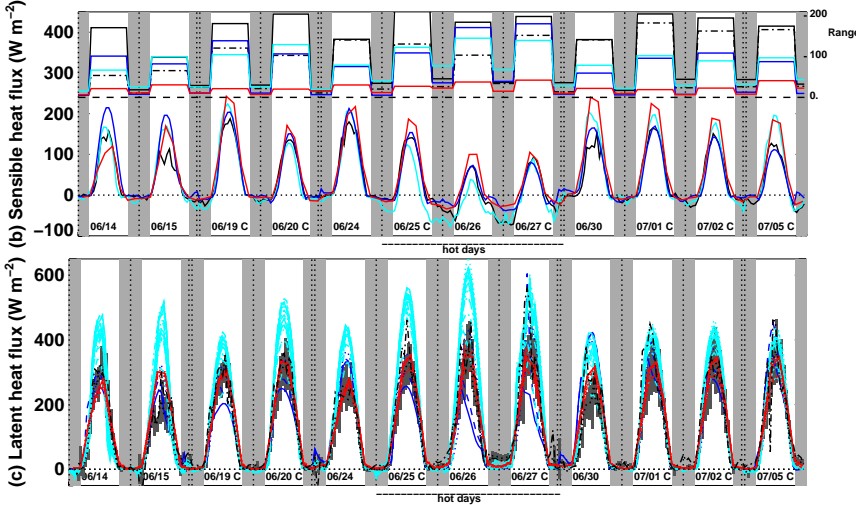

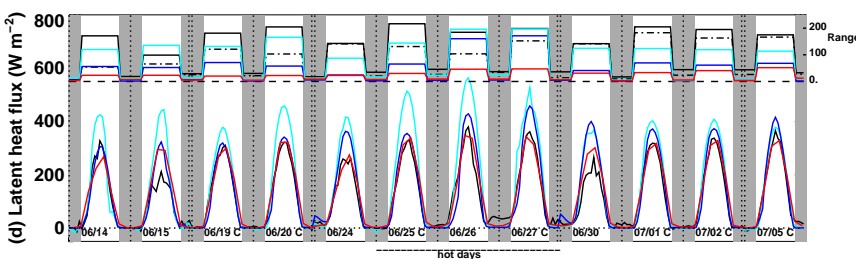

Figure 3: Same as Figure 2 but for (a and b) sensible heat flux and (c and d) latent heat flux. Measurements over several surfaces are indicated in black for the mean and the range (as attested by +/- the standard deviation) shaded in grey in (a), simulated by ARPEGE in dark blue, by AROME in light blue and ECMWF in red. For (a) and (c) the dashed and dot-dashed black lines correspond to the observations over the forest site that are not included in the range spanned by observations indicated by the dark grey shading in (a and c). In (b) and (d), the observed spatial variability is computed either for all the stations (full black line) or for all the stations except the forest sites (dash-dotted black line).



January, 2016

Figure 4: Same as Figure 3 for (a and b) 2m temperature, (c and d) 2m water vapour mixing ratio and (e and f) 10m-wind speed.



January, 2016

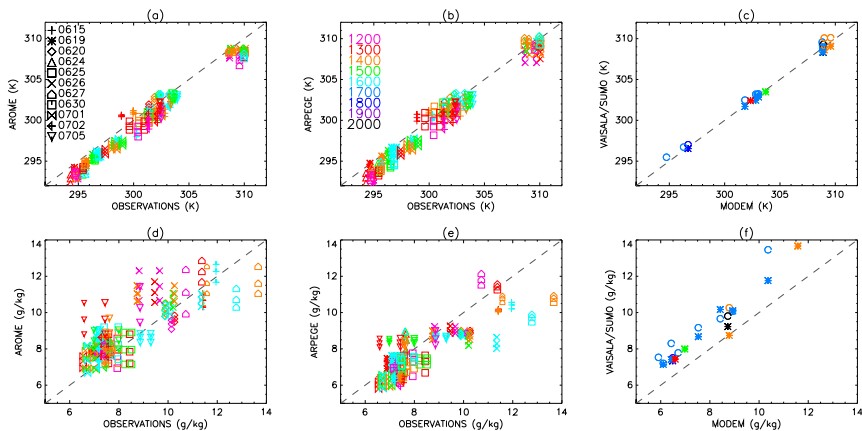

Figure 5: Scatterplot for (a,b,c) the potential temperature and (d, e, f) the water vapour mixing ratio averaged over the first 500 m deep layer: (a and d) AROME values versus the observed values, (b and e) ARPEGE values versus the observation values and (c and f) values obtained from the Vaisala and the SUMO profiles versus the values obtained from the MODEM profiles. Symbols vary from one day to the other and color from one time to the other (see legend).





January, 2016

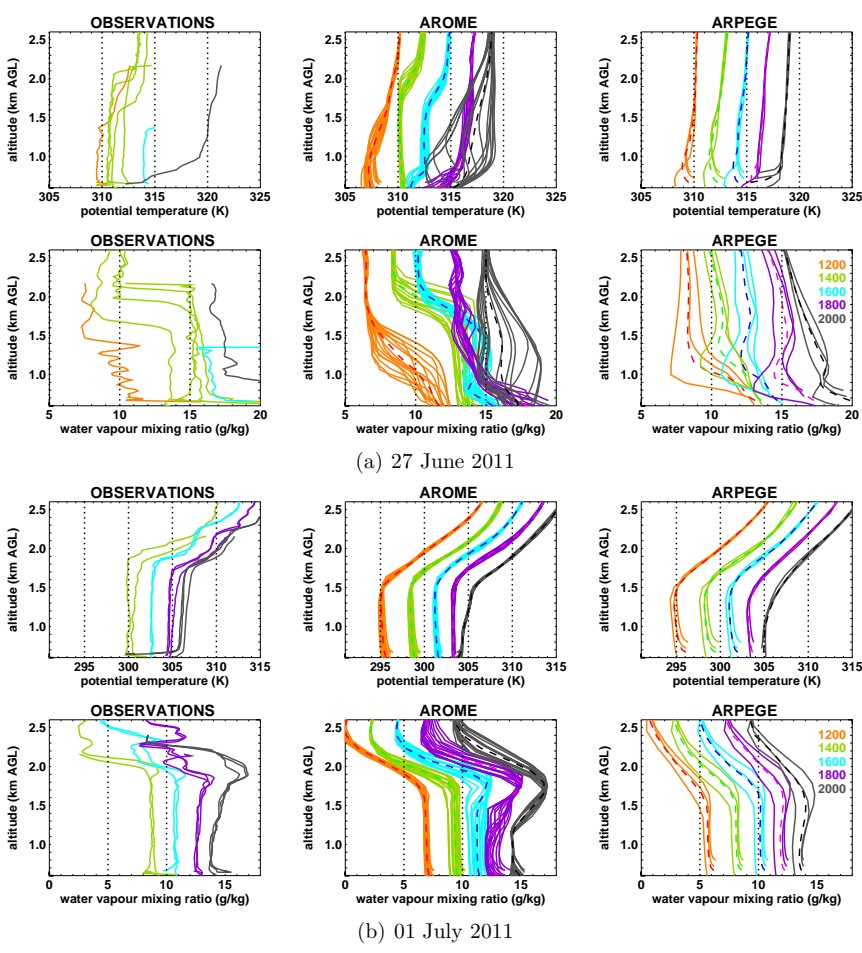

Figure 6: Vertical profiles of potential temperature and water vapour mixing ratio for observations (left panels), AROME (middle panels) and ARPEGE (right panels) for two days the 27 June 2011 (upper panels) and the 01 July 2011 (lower panels). For visibility purposes, the vertical profiles are offset by adding 2K or 2 g/kg every two hours from 1400 to 2000.

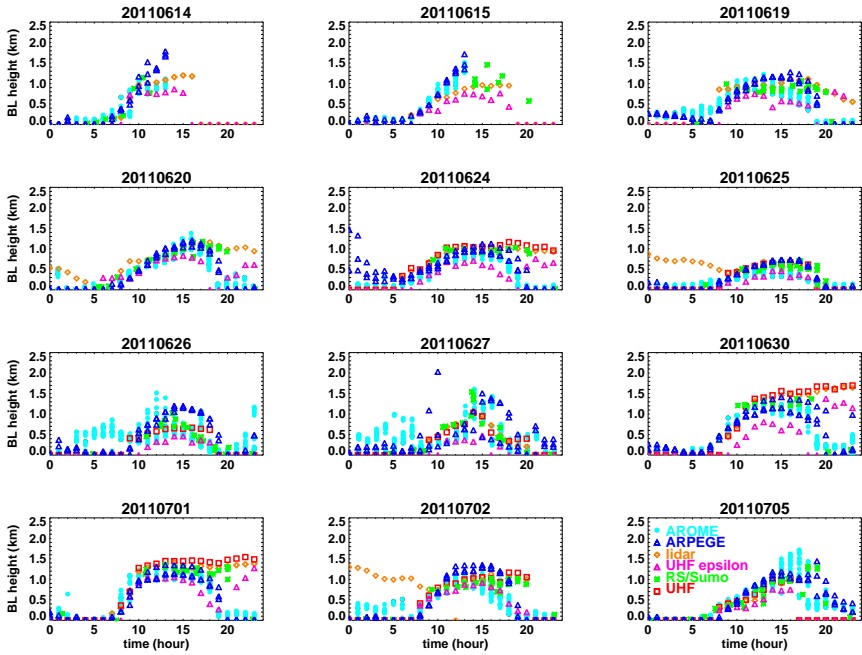

Figure 7: Time series of boundary-layer height for each IOP observed by aerosol lidar(orange diamonds), UHF (from reflectivity in red squares and from the dissipation in pink triangles), radiosoundings or SUMO profiles (green stars) or simulated by ARPEGE (blue triangles) or AROME (cyan full circles). As indicated in the text, no value is drawn from ARPEGE and AROME after 1400 UTC on 14 and 15 June as the existence of clouds induce that the boundary-layer height diagnostic depicts in fact the top of the shallow clouds.



January, 2016

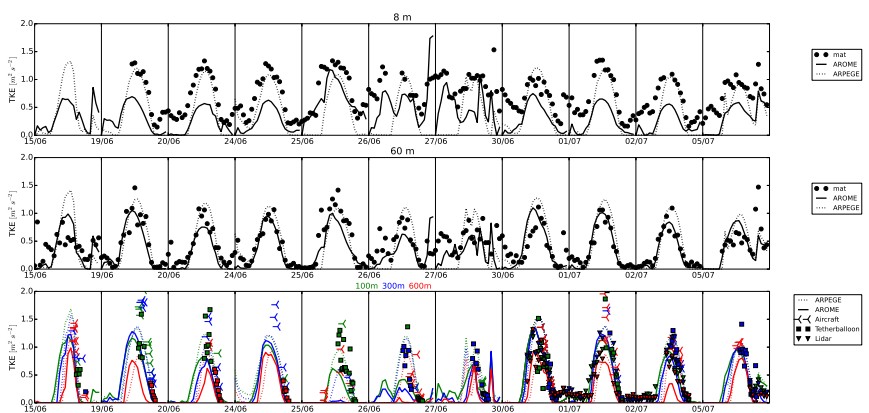

Figure 8: Time series of turbulent kinetic energy observed (in symbols) or simulated by AROME (full line) and ARPEGE (dotted line) at (a) 8m above ground level, (b) 60m above ground level and (c) 100m, 300m and 600m above ground level for the different IOPs from 15 June to 5 July.





January, 2016

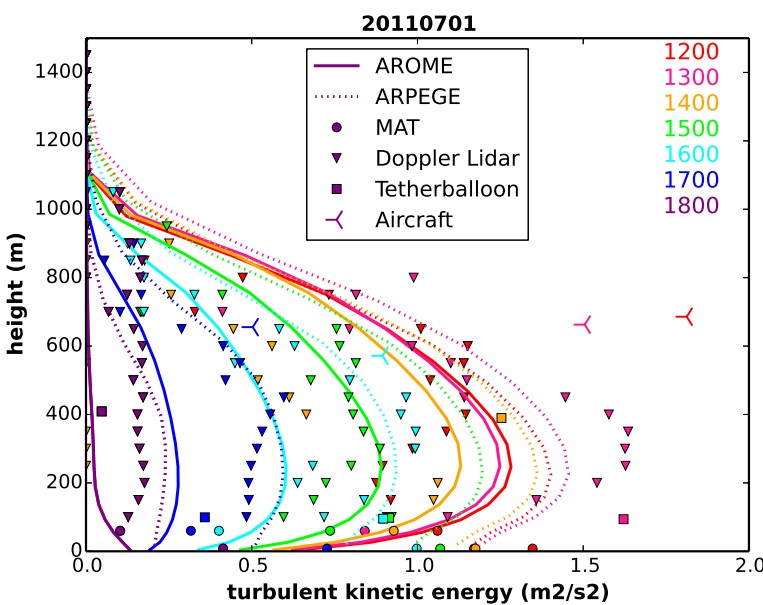

Figure 9: Vertical profiles of the turbulent kinetic energy modelled by AROME (full lines) and ARPEGE (dotted lines) from 1200 to 1800 UTC (see legend), when available, observations are overplotted.





January, 2016

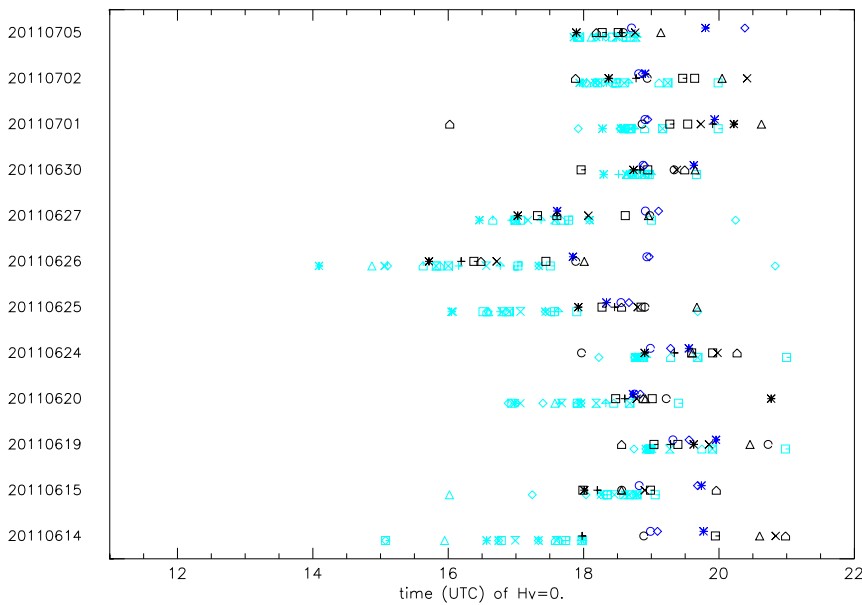

Figure 10: Time (on the x-axis) when the virtual temperature flux goes negative for surface stations observations (black symbols), the ARPEGE grid-points (blue symbols) or the AROME grid-points (cyan symbols) for each IOP plotted on the y-axis.



January, 2016

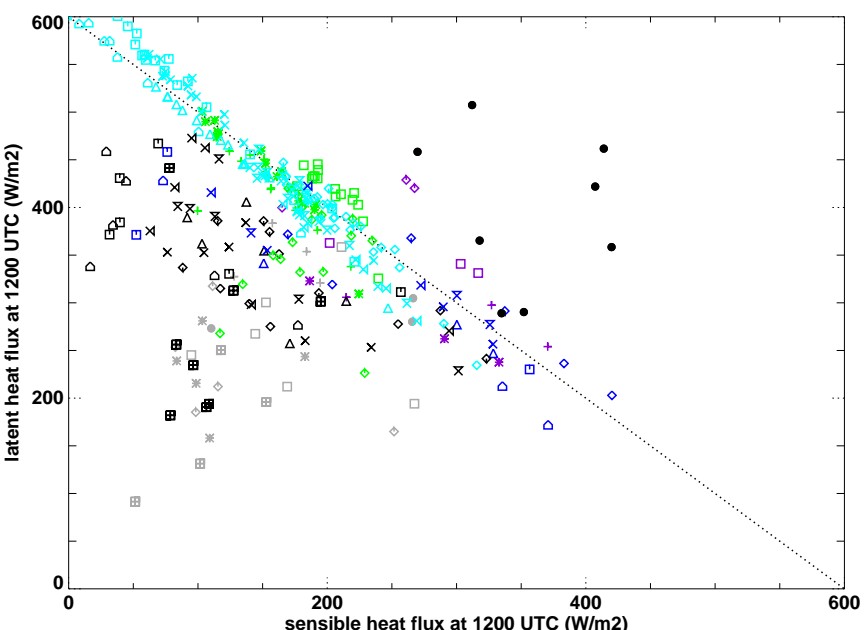

Supplementary Figure 1: Latent heat flux versus Sensible heat flux at 1200 UTC in observations (in black for clear days and grey for cloudy days; the dots correspond to the observations over the forest, while the crossed squares correspond to the observations at 60m in the 60m-tower) and models (AROME in cyan for clear days and green for cloudy days and ARPEGE in blue for clear days and purple for cloudy days). One symbol is plotted for each IOP.




January, 2016

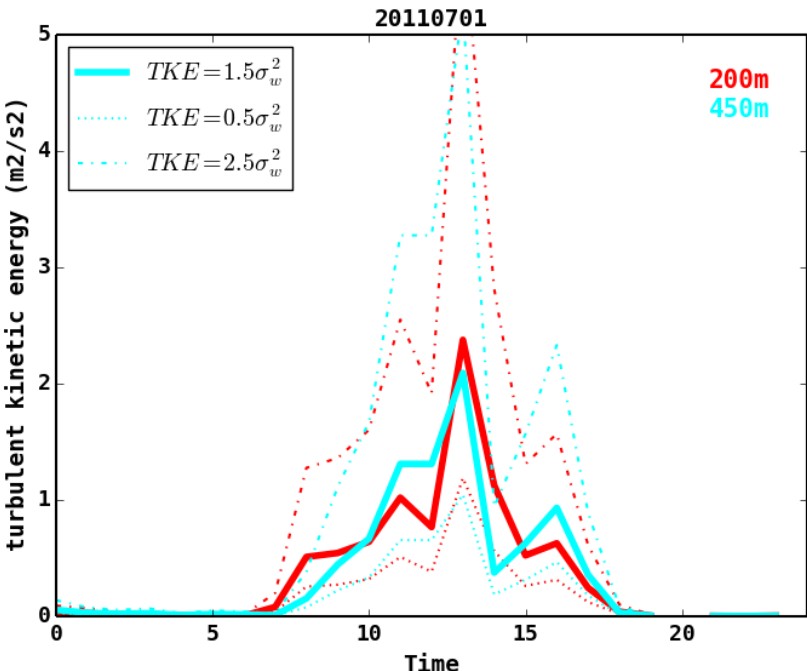

Supplementary Figure 2: Time evolution of the tke derived from $\overline{w'^2}$ lidar observations using various assumptions for A ($A = 0.6$ in dotted lines, $A = 1$ in full lines and $A = 3$ in dashed lines).