# Peer review of "Boundary-layer turbulent processes and mesoscale variability represented by Numerical Weather Prediction models during the BLLAST campaign"

_Atmospheric Chemistry and Physics, 2015_

## Referee Comment (RC1) · Anonymous Referee #1 · 8 Mar 2016

Review of acp-2015-1042 After careful reading, my general impression is that the manuscript contains relevant and sound scientific findings as result of a massive analysis work, and deserves publication. It is a pity, though, that the exposure of such a wealth of results is rather poor. The reading is hard and fragmented, with too many inaccuracies and repetitions. The style needs improvement before the paper can be accepted for publication. To my opinion, the figures are simply not to the ACP standard and require a complete rethinking, not only for publication but even for review. I have struggled to get useful information out of the figures in their current format. I leave the editor the decision if they can be accepted as they are. The structure and 'paragraphing' used for the discussion of the results, on the other hand, seems appropriate, but

the text there also needs editing and proof reading. My suggestion to the editor is, therefore, to ask the authors to amend the manuscript following the suggestions given above and below, and have a second round of review about the scientific content once the necessary changes have been made.

Editing (some...)

Abstract. In general the abstract looks too fragmented. Please revise and try to make more concise. The results portion is too detailed for an abstract. Line 13. Three 'operational wheatear forecasting' models Line 16-17. Arome and ARPEGE are applied over France with a grid-size of 2.5 and 10km respectively, while ECMWF is a global model with 16 km grid-size. Line 19. Representation? Do you mean 'evaluate the models against measured fields'? Line 21 twelve Line 24 evaluated 'by using/against/...' Line 26 variability (or variables???) of cloud cover, temp, boundary layer depth (remove one-day to the next, is not necessary)

Introduction. The opening sentence needs improving and some references to your statements.

Throughout the text, please choose a tense and stick with it (see 'use' at line 27 and 'used' at line 32, etc.).

Not sure I understand the meaning of 'representation' used rather often.

Line 22...'those models'...you only evaluate three of them, which are not mentioned in the introduction but should be (see Line 15 of page 3!). Please make sure the introduction contains the case you want to study in a self-contained way. All the necessary info should be in the introduction: motivation, literature results, gaps in the knowledge you are going to fill with your work. Be precise, I don't personally see the need of being general. Give all the elements you have used and that you are going to develop to support your work.

Line 28 '...mesoscale variability' of what?

First two lines of page 3. I don't understand the meaning of the sentence. Can you please clarify? Second line of page 3. '...single-column runs ARE often used as a simplified configuration OF a full 3D simulation ...' . Also, define 'single-column' models.

Line 10 page 3. '...are quite rare compared to...'

Define tke the first time you introduce the turbulent kinetic energy (same for IOP). Remove 'days' after IOP.

Other suggested edits

Page 4, line 7.' ... all surface stations measuring turbulence...'

The first two lines of section 2.3 can be removed, or at least, rephrased. 'Due to the coarse grid spacing...' Page 6, line 15. '...the tke is below .... In the observations...'

Page 6, line 18. '...usually provides an estimate..., based on the vertical gradient of the relative humidity'

Page 7, line 5. '...at a given hour h correspond...'

The paragraph at the beginning of section 3 should be moved to the methodology section Page 11, line 12. '...variables indicating different...'

Page 11, line 26. '... the boundary layer depth estimated by the model with the boundary layer depth estimated by the observations'.

Page 11, line 29. Please provide reference

Page 11, line 30. '...the temporal variability in terms of maximum boundary layer depth from ona day to the other...' is not clear. Do you mean the variability diurnal cycle?

Page 11, line 34. '...the physics of the models respondS ...'

The end of page 8 is a left-over of some copy-paste?

[Figure]

The first sentence of section 3.3 is unnecessary (already said a few times)

In the Appendix the last words sounds strange..' A=3 would be a value too large'.

Table 2. The roughness length is measured in meters

Figures 2. I would suggest to keep only the mean curves and/or to replace the time series with box and whiskers, four for each IOP (obs plus three models) or three is you prefer to plot the bias (obs – mods). Add the legend to all figures if possible, to help the readers.

Figure 6. The choice of colors is unfortunate. Why not blue, red and green for example? The graph is anyway difficult to interpret, please try to make clearer (in the caption please use 'becomes' in place of 'goes').

---

## Referee Comment (RC2) · M. LeMone (Referee) · 9 Mar 2016

Comments on: Boundary-layer turbulent processes and mesoscale variability represented by numerical weather prediction models during the BLLAST campaign, by Couvreux et al.

General comments

I already made several in my overall quick review (attached at the end of this document). However, I think it is very important to focus more on the impacts of the different grids, in terms of what model TKE is most comparable to observations, in terms of shadowing (and its impact on surface fluxes, especially in the evening and early morning), and in terms of resolved boundary layer structures, which can account for an important part of the TKE (w of order of 1 m/s in Ching et al 2014 MWR and LeMone et al. 2013 MWR).

Also, the impact of the different model terrain, particularly on heterogeneity at night. Acevedo and Fitzjarrald and LeMone et al. (2003, JAS) both show terrain plays a role in nighttime horizontal heterogeneity.

I spend a lot of time writing what model variables might be directly comparable to the TKE measured in the atmosphere. This would be unambiguous if all PBL transport were proportional to the local gradient (i.e., don't need mass flux in the PBL schemes) and there are no resolved PBL eddies (possible with large horizontal grid spacing). It starts to get ambiguous when you have the resolved eddies (I'd just add their TKE to the subgrid TKE), and when you have mass flux in your EDMF schemes. What I don't know is whether the "MF" in the mass flux scheme is by TKE is completely separate from that in the "TKE" part of the scheme. In my comments, I assumed that it was, i.e., that the model TKE was the sum of the subgrid TKE + MF TKE + resolved-eddy TKE.

Peggy LeMone

Specific Comments:

P1 L26. Should be 24-h forecasts

P2 L1-2: Not sure what you mean here. Do you mean that there were more forests in the model or in reality? You could clarify by being more specific, for example, "related to identifying mixed forest and meadows as "forest" in two grid scales in the model." (If there is too much forest in the model).

P2 L10-11. How about "The model reproduced the range of variables to within an order of magnitude." (This is more compact; you don't need to write that it was analyzed).

P2 L29. Don't need "the" before "Europe"

P2 L29. It is interesting that this model has a warm bias in cold and stable conditions. Don't most models show cold biases under such circumstances?

P3, L3-4. LeMone et al. evaluated PBL schemes and their diagnositics.

L25, work of Acevedo and Fitzjarrald. LeMone et al. (2003, JAS) showed from CASES-97 data and evaluation of results of earlier field programs that the timing of maximum horizontal variability depends on the scale of the terrain. This is because of how long it takes for drainage currents to flow from high points to low points. This timing has to be also affected by "frost hollows" that are sheltered from the wind. This makes LES of limited value unless it has a very large domain with very fine mesh. Just a comment; doesn't need a response. And it also implies an important role of model resolution.

P4, L11-14. I don't understand this sentence, especially the use of the word "punctual," which means "on time", as in "She was punctual" -- she arrived just when we expected her to.

Maybe you should just write that observations of TKE profiles, being made only during field campaigns, are quite rare.

P5, L14, L16. Could you describe "moor" in more detail? Is that a specific kind of vegetation or mix of vegetation?

Figure 1. This figure is extremely hard to read. Need bigger range of color or lighter colors. And maybe larger size.

P 6, L7. Suggest "unique aspect" rather than "specificity."

Section 2.2. Suggest details regarding numerical models in a brief table. This helps the reader refer back to model physics (especially the PBL schemes), grid spacing, run length, beginning of runs, etc.

Table 2. Why not include vegetation type, rather than a lot of the detail here, since readers will know, for example, that "forest" has a larger roughness length and LAI than "grassland," and "forest" has a lower albedo than "grassland." And then you could include a column describing what you consider to be the land cover. Or, if not, at least you could refer back to the table when noting result mismatches due to mis-characterization of vegetation.

Also, it would be instructive to including a four-frame figure showing the terrain contours for the three models and what it really looks like.

A fussy comment: should be "grid points" not "grid-points."

Section 2.2.  Also, did you run the ECMWF model or download output?

As to vertical grid points, you could put them in your profile figure to give the reader an idea of where they are.

Section 2.3
P 8, L8-10.  Are you referring to Lothon et al?  If so, refer to it.

P8 L11.  "Clues as to" rather than "inferences on"?

Need to give conversion from UTC to local time, which is what drives PBL development.

P8, L26.  Replace "an" with "a vertical interpolation"

P8 L30.  Which model?  All three?

Also – why don't you try using some of your observational criteria on the model profiles?   In some sense, you are often comparing apples and oranges rather than apples to apples, since different criteria can give different PBL heights.  (See LeMone et al. 2013 – we very rapidly abandoned the diagnosed values because they were often inconsistent with the model theta profiles).

P9, L4.  Can delete "previous"

As noted in earlier general comments, a look at the paper by Lindsey Bennett et al. (MWR, 2010) might be helpful.

P9, L20-21.  See "also" comment above.

P9, L23-5. Why not apply the different criteria to the model profiles to see how they relate within the model?  (I.e., different criteria give different PBL depths).

P10, L3-13.  Evaluation of TKE in the PBL is hard; and comparing it to TKE in the model is even more challenging.

Averages are probably too short; and aircraft high-pass filtering eliminates important scales.   See Grossman et al. (1992) and Kelly et al. (1992), both in J. Geophys. Research for flux profiles.   In the CBL, you should expect to see large eddies of scale of the order of 1.5-3 times the depth of the CBL; a 5-km cutoff will diminish these eddies significantly.  In fact, use of such a short averaging time (and cutoff) is not consistent with the 30-min averages for surface fluxes, which are designed to capture all the fluxes.

Of course this is for capturing the total TKE. For 16-km horizontal grid spacing, this would represent the TKE from the PBL scheme, plus the TKE associated with the parameterized mass flux. For 2.5-km grid spacing, this would be the TKE from the PBL scheme plus the TKE associated with mass flux, plus the TKE associated with partially-resolved eddies.

It's not surprising that you get larger measured TKE than model TKE simply because you don't include the "MF" part (which is only w). Perhaps one meaningful comparison could be made at mid-PBL when the horizontal TKE is smallest and vertical TKE the largest.

Or you could derive a rough representation of mass flux in the PBL scheme by developing an empirical relationship between mass flux and TKE from the aircraft data (for larger scales). Not sure this would work – the relationship between w in TKE and w in mass flux in EDMF schemes is not clear to me.

A really tough but useful test (but more doable than TKE from the model point of view) would be to compare the moisture flux from the model and from the observations, since the model uses total flux divergences (at least for the two coarse-grid models; you would need to add flux by the resolved eddies in the 2.5-km grid spacing model). Heat flux also – but that is so tightly constrained that it doesn't give you as much information. (We tried this in Tastula et al. QJRMS 2015 or 2016).

It is reassuring that your TKE is typically larger than the model TKE – that is what one would expect, given the above discussion. I'd expect the discrepancy to be even larger if you used averages that included the larger scales.

P11, L 19-20. Ambiguous. I thought these four days had no clouds or a few clouds (and by implication, the other days had more clouds). Suggest rewording, perhaps like this.

Those days correspond to mainly high-pressure fair-weather conditions with no cloud cover, or, for 14, 15, 24, and 30 June, a small amount.

In Figure 2 caption, don't use "range" since range means maximum minus minimum. Suggest "black curve with horizontal standard deviations indicated by error bars" instead of "black curves … shaded in grey" since you already have nighttime shaded in gray and this avoids the use of the word "range"). I put in "horizontal" since I think that is what you mean.

Also, you should replace "variability" with "range," which is the correct label – and you have room for a bigger font, which is important. Print is very marginal in size for readability.

Finally, you need to explain the dashed lines in the figure (they are explained in Figure 3). Also, if you label your points in Table 1 with land use, it would help interpretation here as well as in the text.

P12 L1. Similarly, you don't need the "gray shading that indicates the envelope containing the different surface sites," since (a) it's described in the caption, (b) the "gray shading" is confusing, since the error bars look black on the graph and the gray shows night, and (c) an "envelope" typically describes the range (maximum – minimum).

P12 L3. What does "for a given type" mean? Don't you mean for a given day?

P12 L3. This is correct use of "range." So why not use that instead of "variability" on the right side of figure 2. This word is also shorter, so you can make the letters bigger. (I can't read it easily unless I enlarge the electronic version)

P12 L4. Suggest (no C in Figure) after "cloudy days" since you do not explain what the C means (I thought it meant "cloudy"!). Maybe – if possible, you could include a circle with cloud fraction instead of the C. That way it would be less ambiguous (since both "cloudy" and "clear" start with C.

P12 L7. Either "clear" or "cloud-free" but not "clear-free"

Figure 3.

You should repeat the labels on the plot that you put in Fig. 2; also replace "variability" with a "range" in a larger font. Also label the "hot" days, since you discuss them.

P12, L23-4 "which has similar range as observations above the forest". I am not sure what you mean by this. When you say range, are you referring to range in time, since there is only one curve? If you are referring to the difference between two forest sites in model and observations, should point out that they are not shown in the graph. Again, a vegetation type label would be useful.

P12. L29-30. I THINK you are saying that the model assumes more trees in the grid box than there actually is. That is not captured by "much more surface heterogeneities at this size." Also, reference to Fig. 1 doesn't help since you really can't see much (it might if you improve the figure). If you put surface type in the table, this would help. And perhaps label the points in the figure that you discuss in the text. (I.e., you don't have to label all of them).

P12 L31. The only gray shading I see is the nighttime.

P13 L3. Again, please label the hot days somehow on Figure 3.

P13 L17.  Have you looked into the "coupling constant"?   I.e., the coefficient in the bulk formula used to calculate flux?  We have found it sometimes to be off in the model when compared to the observed value.   This could account for both latent and sensible heat flux being too high, since the solar radiation doesn't look that far off.

P13 L18.  What is "high vegetation"?

P13 L24.  This is a new thought, so should start a new paragraph.  I noted in my earlier set of comments the citation to LeMone et al., which you appear to have in the references but not obviously in the text.   As noted previously, a negative slope in the plot means a constant available energy, not a constant Bowen ratio.   For a constant Bowen ratio:

Bowen ratio = B = SH/LH.  If it were constant, LH = SH/B, which would mean that the slope would be positive, not negative.

P14, L17-19. We found that wind reduced horizontal variability during the night for CASES-97 in LeMone et al. (2003).  I would guess Acevedo and Fitzjarrald did as well for their data; because the BL remains coupled to the ground.   In strong winds, we found theta almost constant at night.  Curious that the model didn't – but then you wouldn't get as much terrain-induced variability with the coarser-grid models.  (Again, would be nice to see what the terrain looks like with the coarse-grid models).

Figure 4 caption:  should note what the double vertical lines are.  Did the rain occur at the same time every day, as the figure implies?

Regarding diurnal cycles for mixing ratio (bottom, P 14).  It does look as though you get the morning and evening maxima at least at some sites (associated with large latent heat flux into a shallow BL).  This is a good marker for the creation of the shallow PBL in the evening locally.  If the terrain is complex, perhaps this happens at different times at different sites.

This feature is strongest for weak winds and strong LH.

End, section 3.2 – yes, mixing ratio is the most difficult!

Figure 5.  Regarding warm and cold biases in the lowest 500 m for the models.  Have you factored in differences in PBL depth?  For example, if the PBL depth were underestimated by the models, the mixing ratio would be greater.  (Of course, horizontal advection – and initial conditions – could also have an effect).

Figure 6. Suggest taking advantage of this figure to show where the lowest grid points are.  One could do this by putting points on one profile for each of the models,

or you could mark grid points in a three columns within one of the frames – (top right figure would be excellent for this).

For 27 June, I am intrigued by the large horizontal variability even though the skies are clear.  Do you have resolved PBL eddies?   These can affect surface fluxes, and especially humidity and wind (also temperature, depending on PBL scheme).  On strong-wind days you could be getting model rolls as well as observed rolls, which are associated with strong horizontal changes (see e.g., Weckwerth et al. MWR, 1996).  Also Ching et al. (2014, MWR) and references therein.

P16L11.  "Mesoscale circulation" very vague.  It would be good to give a scale and perhaps a likely cause.  Do you mean terrain-induced circulations?   Or something larger in scale?

In Fig. 7, label the hot days.

It would be more meaningful to compare similarly-diagnosed PBL depths from observations and model.  And to compare differently diagnosed PBLs internal to the model and internal to the observations.

 P16, bottom.  Have you an idea what causes different types of morning PBL growth?  Subsidence?  Strength of inversion?   And as you note, different criteria can give you different PBL depths.  One you might mention is RH max, which will give you the top of the PBL in the absence of cumulus clouds, but will give you cloud base in the presence of cumulus clouds.  (Curiously, we have found that PBL turbulence statistics scale well with cloud base, but it can be argued that the true PBL depth is somewhere in the cumulus cloud layer.)

P17, L1.  Isn't the top of the stable layer and the top of the inversion layer the same thing?

P17, L7.  Better prediction for AROME makes sense to me if you include shading, which could be a factor in decreasing surface buoyancy flux, especially near sunset.  Do you?

P17 L16.  "unique feature" rather than "specificity."

Figure 8.  Bigger font on right side.  I can barely read the labels in my printed version – I'm working off my computer screen.

P17L30-31.  Measured ON the evening … and IS reproduced.

P17, bottom to P 18, top.  I hadn't even thought of slope winds – it's very hard to expect a 1:1 correspondence of measured to model TKE even for a horizontally-homogeneous area – but in complex terrain, it's even more unlikely to expect "agreement" except very roughly.

P 18.  How do model and observed TKE compare if you count the TKE associated with resolved PBL eddies in AROME?  And including some representation of the mass flux associated with the PBL scheme?

P18 L7, L14-15 While the difference in height might be a factor (at the lowest level), you would expect more parameterized TKE in ARPEGE compared to AROME because the PBL scheme in ARPEGE has to do almost all the transport (because the resolved eddies grow much more slowly for 10-km grid than for 2.5-km grid), see Ching et al. 2014, also LeMone et al. 2013).  If it's an EDMF scheme, it should account for all the TKE.  (Again one has to include somehow the mass flux in the TKE estimate).

P18, L19-20.  Resolved PBL eddies grow during the day until saturation is reached. It could be that horizontally-averaged model TKE starts to go down as the resolved eddies grow.

Though I am obviously skeptical that you will even achieve exact agreement, it is encouraging that the trends are similar.

P19L5.  "physical processes … are small "..  You mean terms in the TKE equation are small?  If you don't want to write out the equation, you could write something like

"Most of the terms in the TKE equation, -- buoyancy production, shear production, dissipation – are small."  ?

Or are all the terms small?

P19L15.  "where the height of the reflectivity gradient decreases with time …"?

P19L25-6.  This makes sense, since dissipation and TKE are closely related.

P20 paragraph 1.  The earlier time at which sensible heat flux goes negative at the surface is consistent with large latent heat fluxes.  This makes sense both from the point of view that more of the total energy is going into LH.  But it also means that the buoyancy flux remains positive after the sensible heat flux goes negative.  I would guess that the time when the buoyancy flux goes negative is also earlier for AROME, and this would be more directly related to turbulence generation than sensible heat flux.  It would be good to see what a plot similar to Figure 10 for buoyancy flux looks like.

P20 paragraph 2.Because of the large latent heat fluxes, it might be useful to normalize thing in terms of buoyancy flux rather than sensible heat flux.

P20L32-P21L1,  suggest ..

Models and observations produce lower sensible heat fluxes, higher temperatures, stronger winds, and weaker TKE than (what? For the other days?)

P21 L22-3. From P7, L12-14, I thought that ARPEGE had the same PBL scheme as AROME. This sentence implies there is no "MF" in the ARPEGE PBL scheme. This could be clarified by listing the PBL physics schemes in a table and describing them more carefully. If there is no "MF" in the ARPEGE scheme, then the TKE should be pretty comparable to the total (no high-pass filtering) TKE. (Though I would expect some discrepancy since pure TKE schemes don't really represent what is going on in the CBL). Please clarify.

As noted earlier, the parameterized TKE in an EDMF scheme should be smaller than measured, particularly for fine-mesh model runs (because of the contribution of resolved eddies to the TKE).

P21, end of 2nd paragraph. Estimation of some terms in the TKE budget might be simpler than the estimation of the TKE, at least in terms of direct comparison of model with observations, for reasons discussed earlier.

Supplementary Figure 2. Is this discussed?

P10, bottom. You refer to an Appendix here (which isn't part of the paper). Perhaps you should just refer to supplementary figure 2? Was there an appendix?

Earlier Review:

General comments

The paper is interesting and I think will be in a publishable form with some modifications. I include here only some major thoughts (in no particular order).

1. The SH-vs-LH graphs having a slope close to -1 doesn't indicate a constant Bowen ratio – quite the opposite, it shows horizontal variation in Bowen ratio. Rather, it shows a constant available energy. (I.e., + SH = constant) This is discussed for CASES-97 in LeMone et al. 2003 (J. Hydromet, choosing the averaging interval.) and discussed as a function of soil moisture using both observations and a land-surface model in LeMone et al. (2007). It is nice to see someone exploring this.
2. When discussing horizontal heterogeneity, terrain plays a big role. This it would be good for the authors to show maps of the terrain used in the three NWP models.
   a. This is true, as the authors recognize, because of the presence of mountain-valley circulations of tall types. The different terrains will produce different circulations.

b. This is also true for horizontal variability. Although one gets downslope drainage winds even with gentle terrain, more extreme terrain probably has more cold-air pooling. So there might be less horizontal variability smoothed terrain.

3. When discussing TKE, the measurements will inevitably include the impact of large eddies (horizontal wavelength between 1.5 and 3 times the depth of the PBL, roughly). These are likely partially resolved in AROME, and they tend to grow in models under convective conditions, faster with smaller grid spacing. Comparison to TKE in the other two models is in some sense more realistic from this point of view, since nearly all TKE will be parameterized. For fine grid spacing (< ~4 km), the interaction between this resolved eddies and PBL schemes can exaggerate local variations in TKE (see Ching et al. Monthly Weather Review 2014 and references therein). Other issues:

   a. TKE is mostly horizontal near the surface, especially for eddies extending through the PBL (for which w is very small; from mass-continuity equation).

   b. Large eddies travel roughly at the mean speed of the wind through their depth (i.e., the boundary layer). Thus if one filters according to scale, the scale should be defined not be the wind at the level of the measurement, but by the mean PBL wind.

   c. A philosophical point (discussed in Ching et al.) is the in the "gray zone" or "terra incognita" the PBL scheme should account for all the TKE in the PBL, which for fine-grid models means several grid points horizontally, and there should be no large PBL eddies (convective rolls or cells). (This is the purists' view; the semi-resolved eddies have been useful in storm initiation or propagation – because large eddies, especially rolls, have been shown to play a role in storm propagation and evolution.). One way to look at this is by considering the buoyancy-flux profile. It should be continuous from the surface (where its value is determined by a land-surface model) up through the PBL. If one does time- or space-filtering that is too fine, the fluxes above the surface are too small. I gather from the discussion that the authors were wrestling with this.

4. The authors should look at the paper by Lindsey Bennett et al. (MWR, 2010) regarding estimates from different instrumentation, in addition to the LeMone et al. and Grimsdell and Angevine work cited.

5. Figure 10. Should look at the time when the virtual-temperature flux goes negative in the afternoon; or, similarly, how this time relates to latent heat flux (and hence vegetation type and soil moisture). If I recall correctly, the sensible heat flux went negative earlier where there was large latent heat flux, based on CASES-97 data. Which meant more variability in this time both spatially and from day to day for sensible heat flux than for virtual temperature flux.

General editorial comments (more detail later)

1. The figures are impossible to study in printed form.  I am reviewing the paper with the figures enlarged on the screen.  The labels on the right hand side need to be larger, and "range" might be a better label than "variability." Also it would be helpful to the reader to label the "hot" days referred to in the text.  And have labels on all the figures to make it easier for the reader.
2. It would help in the profile figure (6) to figures to have grid points on the curves for each model – for one curve for each model.  Also, might consider plotting the average profile for each model and time.  And finally, might consider offsetting the soundings by adding a few degrees for each time interval.  The last might not be practical.  (You could stretch the horizontal axis and only have one altitude label).
3. It would be useful to have a table describing the properties of each model (horizontal and vertical grid spacing, PBL scheme, etc.) as well as model-run length and initiation time and data used to initialize the model.  Also, how the land-surface properties were initialized (often there is a long-spinup).

Peggy LeMone 25 January 2016

---

## Author Comment (AC1) · 20 Jun 2016

**Answer to the reviewer 1 about the manuscript entitled : 'Boundary-layer turbulent processes and mesoscale variability represented by Numerical Weather Prediction models during the BLLAST campaign' by F. Couvreux et al.:**

First, we wish to thank the reviewer for her careful and very detailed review. Below is our response (in blue) to the comments on a point-by-point basis. Reference to how we plan to modify the text is indicated in italic.

General comments :

I already made several in my overall quick review.

The answer to this quick review is attached at the end of this document.

However, I think it is very important to focus more on the impacts of the different grids, in terms of what model TKE is most comparable to observations, in terms of shadowing (and its impact on surface fluxes, especially in the evening and early morning), and in terms of resolved boundary layer structures, which can account for an important part of the TKE (w of order of 1 m/s in Ching et al 2014 MWR and LeMone et al. 2013 MWR).

We have look at the AROME forecasts and verified that no spurious convectively induced secondary circulations were present in those forecasts (horizontal maps of the temperature at different vertical levels are available on the BLLAST website: http://boc.sedoo.fr/nwp/lammodel/arome). Indeed, the effective resolution of the AROME model is around 9 Δx as shown in Ricard et al (2013). Note that simulations with Meso-NH, a research model, that has a smaller effective resolution, more on the order of 3-5 Δx do shown spurious circulations at 2km resolution. For ARPEGE and ECMWF, with horizontal resolution greater than 10 km, the boundary-layer structures are entirely parameterized. For any of those 3 models, the resolved vertical velocity is very small. So here, the resolved boundary layer structures are not an issue. However, we modify the text and now reference the above papers, to stress that in other situations resolved spurious boundary layer structures can be an issue.

[Figure]

Figure 0: horizontal map of potential temperature at 1200 (left figure) and 1600 (right figure) for the 1st July 2011

Also, the impact of the different model terrain, particularly on heterogeneity at night. Acevedo and Fitzjarrald and LeMone et al. (2003, JAS) both show terrain plays a role in nighttime horizontal heterogeneity.

As you suggested, we have now included a figure showing the terrain represented in the different models as well as the real terrain. We also have included more discussion relative to the role of terrain on night heterogeneity (see response to detailed comments below).

I spend a lot of time writing what model variables might be directly comparable to the TKE measured in the atmosphere. This would be unambiguous if all PBL transport were proportional to the local gradient (i.e., don't need mass flux in the PBL schemes) and there are no resolved PBL eddies (possible with large horizontal grid spacing). It starts to get ambiguous when you have the resolved eddies (I'd just add their TKE to the subgrid TKE), and when you have mass flux in your EDMF schemes. What I don't know is whether the "MF" in the mass flux scheme is by TKE is

completely separate from that in the "TKE" part of the scheme. In my comments, I assumed that it was, i.e., that the model TKE was the sum of the subgrid TKE + MF TKE + resolved-eddy TKE.

As said previously, there is no resolved vertical eddies with a 2.5km resolution in AROME. The mass-flux scheme is a more important issue that we partly discarded. The budget analysis of this contribution indicates that the mass-flux scheme provides a small contribution close to the surface, less than 10% of the total tke but a stronger one in the middle of the convective boundary layer where it reaches 20-25%. We therefore revised the comparison by including the mass-flux scheme contribution to the total tke and modified the text accordingly. The figures below present the time evolution of the total turbulent kinetic energy (subgrid turbulence scheme + mass-flux scheme + resolved eddies) at two different altitudes for two different days. We can clearly see that the resolved eddies contributions is null for the 16 different points (dash-dotted lines). The mass-flux scheme contribution is smaller than the subgrid turbulence scheme and accounts for around 10% of the total turbulent kinetic energy at 60m and around 20% in the middle of the boundary layer (illustrated here at 250m).

Figure 1: time evolution of the turbulent kinetic energy (total in full line, subgrid turbulent scheme contribution (dotted line), mass-flux scheme contribution (dashed line) and resolved eddies (dash-dotted line) for the 19 June 2011 on the upper panels and the 01 July 2011 on the lower panels and at 60m on the left panels and at 250m on the right panels

Specific Comments:
P1 L26. Should be 24-h forecasts
Done

P2 L1-2 (=P1 L22): Not sure what you mean here. Do you mean that there were more forests in the model or in reality? You could clarify by being more specific, for example, "related to identifying mixed forest and meadows as "forest" in two grid scales in the model." (If there is too much forest in the model).

This is now better explained in the text. Here, we meant that in ARPEGE there is an overestimation of the sensible heat fluxes as if the grid boxes were entirely covered by forest while the analysis of the Land-Use map indicates that only part of the 10-km grid box (less than 25%) is covered by forest. We removed the words 'over-predominance of forest' from the abstract.

P2 L10-11. How about "The model reproduced the range of variables to within an order of magnitude." (This is more compact; you don't need to write that it was analyzed).
We have made the proposed change.

P2 L29. Don't need "the" before "Europe"
Done

P2 L29. It is interesting that this model has a warm bias in cold and stable conditions. Don't most models show cold biases under such circumstances?
In the study of Atlaskin and Vihma (2012), they tested four models, among which AROME and ECMWF. They showed a positive bias for the 2-m temperature under very low temperature (T<-10°C; a negative bias is observed for less cold temperature at night). We modified the text to make it clearer: '*They focused on the representation of very stable conditions at very low temperature (<-10°C) in northern Europe and showed a systematic positive bias for the 2-m temperature due to an underestimation of the stratification during the coldest nights characterized by very stable conditions.*'

P3, L3-4. LeMone et al. evaluated PBL schemes and their diagnositics.
We changed this sentence to '*LeMone et al (2003) used CASES-97 observations to evaluate boundary-layer schemes and their diagnostics based on mesoscale model simulations'*.

L25, work of Acevedo and Fitzjarrald. LeMone et al. (2003, JAS) showed from CASES-97 data and evaluation of results of earlier field programs that the timing of maximum horizontal variability depends on the scale of the terrain. This is because of how long it takes for drainage currents to flow from high points to low points. This timing has to be also affected by "frost hollows" that are sheltered from the wind. This makes LES of limited value unless it has a very large domain with very fine mesh. Just a comment; doesn't need a response. And it also implies an important role of model resolution.
We added a sentence in the text highlighting the importance of model resolution : '*This highlights the important role of fine resolution in order to get the right orography in the model.* '. There is also now a new figure that includes both the orography and the modelled terrain. See response to major comment n°2.

P4, L11-14. I don't understand this sentence, especially the use of the word "punctual," which means "on time", as in "She was punctual" -- she arrived just when we expected her to. Maybe you should just write that observations of TKE profiles, being made only during field campaigns, are quite rare.
Thanks for your suggestion we included the sentence following your proposition : «*For example, observations of tke profiles, being made only during field campaigns are quite rare, therefore the boundary layer parametrization based on a prognostic equation of the turbulent kinetic energy, which has been shown to perform better than first-order scheme (Holt and Raman, 1988), has only been evaluated via comparison with LES results (Cuxart et al, 2006 for instance)."*

P5, L14, L16. Could you describe "moor" in more detail? Is that a specific kind of vegetation or mix of vegetation?
Indeed, this is a specific kind of vegetation. Moor is an area of open wasteland, often overgrown with grass and heath. We have included this information after the first use of this term.

Figure 1. This figure is extremely hard to read. Need bigger range of color or lighter colors. And maybe larger size.

We have enlarged this figure and hope that now it is clearer.

P 6, L7. Suggest "unique aspect" rather than "specificity."
Done

Section 2.2. Suggest details regarding numerical models in a brief table. This helps the reader refer back to model physics (especially the PBL schemes), grid spacing, run length, beginning of runs, etc.
This was already added in the second version of the submission material after your first quick review.

Table 2. Why not include vegetation type, rather than a lot of the detail here, since readers will know, for example, that "forest" has a larger roughness length and LAI than "grassland," and "forest" has a lower albedo than "grassland." And then you could include a column describing what you consider to be the land cover. Or, if not, at least you could refer back to the table when noting result mismatches due to mischaracterization of vegetation. Also, it would be instructive to including a four-frame figure showing the terrain contours for the three models and what it really looks like. A fussy comment: should be "grid points" not "grid-points."
We have included the dominant vegetation type in a supplementary column of Figure 2 in order to help the reader's interpretation. However, we decided to keep the surface characteristics (albedo, roughness length, vegetation fraction,..) of the different points as this corresponds to the values that are used in the computation of the energy budget. For AROME and ARPEGE, they have been calculated taking into account the subgrid variability of the land use as explained in Giard and Bazile (2001). A four-frame figure showing the terrain contours for the three models and in the real world has now been included in the manuscript (new figure 2).
Throughout the text, 'grid-points' was changed into 'grid points'.

Section 2.2. Also, did you run the ECMWF model or download output? As to vertical grid points, you could put them in your profile figure to give the reader an idea of where they are.
We did not run the ECMWF model, it was run operationally by the European Center. We retrieved the model outputs from the ECMWF archive and analyzed them. There is no figure showing vertical profiles of ECMWF runs but the information concerning the vertical resolution is already included in Table 2.

Section 2.3
P 8, L8-10. Are you referring to Lothon et al? If so, refer to it.
We have modified the sentence to be more explicit : « *A large variability of surface fluxes exists among the sites (Fig 1) at scales smaller than 2.5x2.5 km², which corresponds to the size of a grid box in AROME (see for example the differences between the moor and the corn sites, or the grass and the wheat sites) that are mainly due to surface cover; this was also shown in Lothon et al (2014)*"

P8 L11. "Clues as to" rather than "inferences on"?
Done

Need to give conversion from UTC to local time, which is what drives PBL development. At this location (Lannemezan, France; lon=0.38°E) the longitude is very close to the 0° Greenwich meridian. So, UTC time is very close (~2min) to solar time. However, in France the local time is postponed so that 1400 LT is equal to 1200 UTC time and 1200 solar time. So we have kept the UTC time in the paper but we also indicated that this is very close to solar time. We have included the following text:
*'… note here that UTC time is the same as solar time as very close to the Greenwich meridian'*

P8, L26. Replace "an" with "a vertical interpolation"
Done

P8 L30. Which model? All three? Also – why don't you try using some of your observational criteria on the model profiles? In some sense, you are often comparing apples and oranges rather than apples to apples, since different criteria can give different PBL heights. (See LeMone et al. 2013 – we very rapidly abandoned the diagnosed values because they were often inconsistent with the model theta profiles).

Here we only analysed ARPEGE and AROME models as the ECMWF finer available time sampling (3 hours) was too coarse to investigate the temporal evolution. It is not always straightforward to use the same boundary-layer diagnostics for observations and models. Indeed, in observations we use different types of diagnostics derived either from the UHF (two different diagnostics), from an aerosol lidar (one diagnostic) or from thermodynamical profiles (four diagnostics). As you suggested, we applied to the models the diagnostics based on thermodynamical profiles and we now state in the text the results of the comparison of those diagnostics to the model diagnostic (based on *tke*). However, during the afternoon transition, the diagnostics based on thermodynamical vertical profiles sometimes depicts the top of the residual layer rather than the top of the still convectively active shallower layer. In the figures below (illustrated for four IOP days), we compare the diagnostic computed online based on the vertical profiles of the turbulent kinetic energy in black/grey for AROME/ARPEGE with the diagnostic based on the virtual potential temperature in green/blue for AROME/ARPEGE.

[Figure]

Figure 2: time evolution of boundary-layer height diagnosed by the model (based on tke) for AROME (black) and ARPEGE (grey) or diagnosed from the vertical profile of the virtual potential temperature for AROME (green) and ARPEGE (blue) for the 16 points.

There is consistency between both diagnostics for most of the models with however some discrepancy for some times (in particular during the afternoon transition). We therefore decided to keep the model diagnostics (discarding however the time where it is not relevant due to the presence of shallow clouds, this diagnostic depicts the top of the shallow clouds : two hours for the 15 June) as well as time where strong shear induces a decoupling between the boundary-layer and the tke profiles (morning of the 27 June) as illustrated in LeMone  et al (2013) for the shear case. Eventually, also not that with observations we derive different diagnostics with the idea to analyse what each diagnostic depicts in particular during the transition.

P9, L4. Can delete "previous"

Done

As noted in earlier general comments, a look at the paper by Lindsey Bennett et al. (MWR, 2010) might be helpful.

After your first quick review, we included a reference to this paper in this section. This paper is quoted twice in page 6 :'A *comparison of different boundary-layer depths derived from various instruments has been presented in Bennett et al (2010).*'  and '*The decrease of the boundary-layer depth in the afternoon transition is a delicate process and in practice, its estimation is sensitive to the criteria used to derive the boundary-layer depth as already shown by Angevine and Grimsdell (2002) and Bennett et al (2010).'*

P9, L23-5. Why not apply the different criteria to the model profiles to see how they relate within the model? (I.e., different criteria give different PBL depths).

See response to comment (P8 L30) above.

P10, L3-13. Evaluation of TKE in the PBL is hard; and comparing it to TKE in the model is even more challenging. Averages are probably too short; and aircraft high-pass filtering eliminates important scales. See Grossman et al. (1992) and Kelly et al. (1992), both in J. Geophys. Research for flux profiles. In the CBL, you should expect to see large eddies of scale of the order of 1.5-3 times the depth of the CBL; a 5-km cutoff will diminish these eddies significantly. In fact, use of such a short averaging time (and cutoff) is not consistent with the 30-min averages for surface fluxes, which are designed to capture all the fluxes.

The 5-km cutoff is what is usually used in the program computing fluxes from the high-frequency aircraft data. We analysed the sensitivity of turbulent fluxes to the choice of this cutoff length for BLLAST and other field campaigns (AMMA & HYMEX) and found that increasing this cutoff length did not strongly modified the fluxes estimations. However, as expected the computation of the variance is decreased by the use of the 5-km cutoff as illustrated in the figure below, and this effect is stronger for the variance of the horizontal wind compared to the variance of the vertical wind.

[Figure]

Figure 3: Comparison of the variance computed from filtered signal (x-axis) or raw data (y-axis) for (left figure) vertical velocity variance, (middle figure) zonal wind variance and (right figure) meridional wind variance.

For the turbulent kinetic energy, the 5-km cutoff induced a reduction of 20-22% as shown in the figure below:

This is now commented in the text as :
*'...; this is the current treatment used for flux computation, it however induces an underestimation of the tke of about 20%'*
The wind during BLLAST is relatively weak, typically from 1 to 3 m/s so a 30 min average correspond to 30 min ~ 2-5 km and is therefore consistent with a 5km cutoff length. Eventually, during BLLAST, the boundary-layer height was usually around 1 km so the scale of the large eddies should be broadly resolved with such measurements. The segments used to compute the turbulent kinetic energy used for comparison to the models are on average 31km-long and last for 7.5 min (450 s).

[Figure]

Figure 4 :Comparison of the turbulent kinetic energy computed from filtered signal (x-axis) or raw data (y-axis).

Of course this is for capturing the total TKE. For 16-km horizontal grid spacing, this would represent the TKE from the PBL scheme, plus the TKE associated with the parameterized mass flux. For 2.5-km grid spacing, this would be the TKE from the PBL scheme plus the TKE associated with mass flux, plus the TKE associated with partially-resolved eddies.
It's not surprising that you get larger measured TKE than model TKE simply because you don't include the "MF" part (which is only w). Perhaps one meaningful comparison could be made at mid-PBL when the horizontal TKE is smallest and vertical TKE the largest.
Or you could derive a rough representation of mass flux in the PBL scheme by developing an empirical relationship between mass flux and TKE from the aircraft data (for larger scales). Not sure this would work – the relationship between w in TKE and w in mass flux in EDMF schemes is not clear to me.
At 2.5km, we checked that there were no resolved eddied and we have added the contribution of the mass-flux scheme which is negligible close to the surface but more significant in the middle of the boundary layer (see also response to general  comment).
A really tough but useful test (but more doable than TKE from the model point of view) would be to compare the moisture flux from the model and from the observations, since the model uses total flux divergences (at least for the two coarse-grid models; you would need to add flux by the resolved eddies in the 2.5-km grid spacing model). Heat flux also – but that is so tightly constrained that it doesn't give you as much information. (We tried this in Tastula et al. QJRMS 2015 or 2016).
It is reassuring that your TKE is typically larger than the model TKE – that is what one would expect, given the above discussion. I'd expect the discrepancy to be even larger if you used averages that included the larger scales.
Unfortunately the moisture flux was not an output of the model so this is a really tougher test that we decided not to carry out.

P11, L 19-20. Ambiguous. I thought these four days had no clouds or a few clouds (and by implication, the other days had more clouds). Suggest rewording, perhaps like this.
Those days correspond to mainly high-pressure fair-weather conditions with no cloud cover, or, for 14, 15, 24, and 30 June, a small amount.
Done, we included your suggestion.

In Figure 2 caption, don't use "range" since range means maximum minus minimum. Suggest "black curve with horizontal standard deviations indicated by error bars" instead of "black curves … shaded in grey" since you already have nighttime shaded in gray and this avoids the use of the word "range"). I put in "horizontal" since I think that is what you mean. Also, you should replace "variability" with "range," which is the correct label – and you have room for a bigger font, which is important. Print is very marginal in size for readability. Finally, you need to explain the dashed lines in the figure (they are explained in Figure 3). Also, if you label your points in Table 1 with land use, it would help interpretation here as well as in the text.
Eventually, following reviewer 2, we decided to simplify this figure and only showed the mean curves. However, to be able to illustrate the over-estimation of two grid points of ARPEGE we have added a figure (now Figure 3) showing the comparison of sensible and latent heat fluxes between all the observation sites and the different points of ARPEGE. In this caption, we have used your proposition ('*black curve for the mean with horizontal standard deviations indicated by error bars*'). 'Range' was already replaced 'variability' with a bigger font following your early review.

P12 L1. Similarly, you don't need the "gray shading that indicates the envelope containing the different surface sites," since (a) it's described in the caption, (b) the "gray shading" is confusing, since the error bars look black on the graph and the gray shows night, and (c) an "envelope" typically describes the range (maximum – minimum).
Now Figure 2 only shows the mean and the range. Figure 3 presents the horizontal variability but vertical error bars are plotted and the gray shading has been removed.

P12 L3. What does "for a given type" mean? Don't you mean for a given day?
We have modified the sentence to  "*this is computed at each time step by the difference between the maximum and the minimum over all the points of either one model or the observations*"

P12 L3. This is correct use of "range." So why not use that instead of "variability" on the right side of figure 2. This word is also shorter, so you can make the letters bigger. (I can't read it easily unless I enlarge the electronic version)
Following your first quick review, we already changed 'variability' into 'range' in the last submitted version. This has been changed for the 3 figures.

P12 L4. Suggest (no C in Figure) after "cloudy days" since you do not explain what the C means (I thought it meant "cloudy"!). Maybe – if possible, you could include a circle with cloud fraction instead of the C. That way it would be less ambiguous (since both "cloudy" and "clear" start with C.
We agree that the 'C' was ambiguous. It has been changed by an empty circle for cloud-free days and a grey triangle for cloudy days. We did not have quantitative observations of cloud fraction so we could not include this information.

P12 L7. Either "clear" or "cloud-free" but not "clear-free" Figure 3.
We have changed clear-free to clear in the text.
You should repeat the labels on the plot that you put in Fig. 2; also replace "variability" with a "range" in a larger font. Also label the "hot" days, since you discuss them.
Following your first quick review, we already labeled the 'hot days' in Figures 2, 3 and 4**.**

P12, L23-4 "which has similar range as observations above the forest". I am not sure what you mean by this. When you say range, are you referring to range in time, since there is only one curve? If you are referring to the difference between two forest sites in model and observations, should point out that they are not shown in the graph. Again, a vegetation type label would be useful.

In fact, we wanted to state that the values predicted by ARPEGE for the high-vegetation grid points are of the same order of magnitude as the observations above the forest. However, these simulated sensible heat fluxes are too large to be representative of a 10km wide grid box over an area which is characterized by much more surface heterogeneity and is far from being entirely covered by forest (cf Fig 1).We changed the text to :

'...However, these simulated sensible heat fluxes are too large to be representative of a 10km wide grid box over an area which cannot be characterized, according to Figure 1, by a uniform forest cover; indeed, there is a large variability of surface covers at scales below 10km '… 'For two ARPEGE points the surface fluxes are similar to measurements over forest, but the satellite data does not indicate a homogeneous forest patch over 10x10km² in this 10x10km² area. '

P12. L29-30. I THINK you are saying that the model assumes more trees in the grid box than there actually is. That is not captured by "much more surface heterogeneities at this size." Also, reference to Fig. 1 doesn't help since you really can't see much (it might if you improve the figure). If you put surface type in the table, this would help. And perhaps label the points in the figure that you discuss in the text. (I.e., you don't have to label all of them).

This was not clear and we modified the text as 'However, these simulated sensible heat fluxes are too large  to be representative of a 10km wide grid box over  an area which cannot be characterized, according to Figure 1, by a uniform  forest cover; indeed, there is a large variability of surface covers at scales below 10km.' The points which are referred to in the text are now labelled in Figure 1. Figure 1 has also been enlarged.

P12 L31. The only gray shading I see is the nighttime.

According to your previous comment, we change the gray envelope into error bars so now there is indeed only gray shading for nighttime.

P13 L3. Again, please label the hot days somehow on Figure 3.

Following your first quick review, we already labeled the 'hot days' in Figures 2, 3 and 4.

P13 L17. Have you looked into the "coupling constant"? I.e., the coefficient in the bulk formula used to calculate flux? We have found it sometimes to be off in the model when compared to the observed value. This could account for both latent and sensible heat flux being too high, since the solar radiation doesn't look that far off.

Ideally to more fully explore the surface energy budget, we should look at the G component as well, but it was not available in the models. In fact, in the different models the coefficient used in the bulk formula used to calculate the flux is not constant but is computed iteratively and is a function of stability so it is tough to look at this 'coupling constant' and we did not do it.

P13 L18. What is "high vegetation"?

ARPEGE uses a criterion to separate 'high vegetation' from 'low vegetation' in term of stomatical resistance and roughness length. However, this information  is not really necessary here. We have removed the term 'high vegetation' in the text.

P13 L24. This is a new thought, so should start a new paragraph.

Thanks for the comment. We started a new paragraph.

I noted in my earlier set of comments the citation to LeMone et al., which you appear to have in the references but not obviously in the text. As noted previously, a negative slope in the plot means a

constant available energy, not a constant Bowen ratio. For a constant Bowen ratio:
Bowen ratio = B = SH/LH. If it were constant, LH = SH/B, which would mean that the slope would be positive, not negative.

After your quick first review, this reference has been included in the text (cf P9 l30: 'Interestingly, when plotting the latent heat fluxes as a function of the sensible heat fluxes at 1200 UTC, the models reproduce the -1 slope related to an almost constant available energy (cf Supplementary Fig 1) in agreement with LeMone et al (2003).')

P14, L17-19. We found that wind reduced horizontal variability during the night for CASES-97 in LeMone et al. (2003). I would guess Acevedo and Fitzjarrald did as well for their data; because the BL remains coupled to the ground. In strong winds, we found theta almost constant at night. Curious that the model didn't – but then you wouldn't get as much terrain-induced variability with the coarser-grid models. (Again, would be nice to see what the terrain looks like with the coarse-grid models).

Concerning the horizontal variability at night we have added these two references: '*The spatial variability in night time temperature among sites is smaller for the hot period; this is probably be due to larger wind speed during this period (as shown in LeMone et al 2003 and Acevedo and Fitzjarrald 2001).*'
We have also included a figure showing the terrain in models and observations (cf answer to the second general comment and new figure 2).

Figure 4 caption: should note what the double vertical lines are. Did the rain occur at the same time every day, as the figure implies? Regarding diurnal cycles for mixing ratio (bottom, P 14). It does look as though you get the morning and evening maxima at least at some sites (associated with large latent heat flux into a shallow BL). This is a good marker for the creation of the shallow PBL in the evening locally. If the terrain is complex, perhaps this happens at different times at different sites. This feature is strongest for weak winds and strong LH.

The double vertical dotted lines indicate interruptions in the days as only the IOP days are plotted and not all the days from 14 June to 2$^{nd}$ of July. This is now added in the caption of new figures 2 and 4. There is no explicit mention of the time of the rain, the rain often occurs at night but not always at the same time. This is also mentioned in the text as :' *the days with precipitation were not IOPs and corresponds therefore to an interruption of time in Figure 4, indicated by the double vertical dotted lines*'.
We have also included a comment regarding the morning and evening maxima: '*Often, observations indicate a morning and evening maxima (e.g. 19 June, 30 June, 01 July, 02 July) associated with large latent heat flux into a shallow boundary layer; this is correctly simulated by the models.*' However, the relationship with the intensity of winds and surface latent heat fluxes is not so obvious.

End, section 3.2 – yes, mixing ratio is the most difficult!
Figure 5. Regarding warm and cold biases in the lowest 500 m for the models. Have you factored in differences in PBL depth? For example, if the PBL depth were underestimated by the models, the mixing ratio would be greater. (Of course, horizontal advection – and initial conditions – could also have an effect).
For AROME and ARPEGE, there is no obvious biases in terms of the PBL depth. Concerning the ECMWF dry bias, we have checked that it is not related to a too high PBL depth either.

Figure 6. Suggest taking advantage of this figure to show where the lowest grid points are. One could do this by putting points on one profile for each of the models, or you could mark grid points in a three columns within one of the frames – (top right figure would be excellent for this).
As you proposed, the vertical grid of the models is indicated by crosses in this figure.

For 27 June, I am intrigued by the large horizontal variability even though the skies are clear. Do you have resolved PBL eddies?

For sure, there is no resolved PBL eddies in the ARPEGE simulation due to the coarse 10km-resolution. We have also checked in the AROME simulation and AROME does not either present resolved PBL eddies (see also response to first general comment). In addition, ARPEGE, as AROME, shows a large horizontal variability for this day. This large variability seems to be related to the synoptic conditions as the 25 June (during the hot period) large wind may have prevented the establishment of the mountain-plain circulation or at least delays it.

These can affect surface fluxes, and especially humidity and wind (also temperature, depending on PBL scheme). On strong-wind days you could be getting model rolls as well as observed rolls, which are associated with strong horizontal changes (see e.g., Weckwerth et al. MWR, 1996). Also Ching et al. (2014, MWR) and references therein.

Thanks for this reference, but we have checked and there is no convectively induced secondary circulations during this day.

P16L11. "Mesoscale circulation" very vague. It would be good to give a scale and perhaps a likely cause. Do you mean terrain-induced circulations? Or something larger in scale?

We have added a scale. In fact, carrying a simulation with the same physics as ARPEGE but at 2.5 km horizontal resolution also reproduces the maximum in the upper part of the boundary layer as shown in the figure below. This indicates that this feature is related to fine scale advection not resolved with a 10-km grid : *'Analysis of the moisture budget indicates that this maximum is mainly related to fine scale advection not resolved at 10 km (not shown).'*

Figure 5: vertical profiles of specific humidity the 1st of July simulated by AROME (2.5km resolution, in blue), ARPEGE (10 km resolution in black) and ARPEGE physics using the 2.5km dynamics of AROME (in red).

[Figure]

In Fig. 7, label the hot days.

The hot days are now labeled in this figure and also the previous figures.

It would be more meaningful to compare similarly-diagnosed PBL depths from observations and model. And to compare differently diagnosed PBLs internal to the model and internal to the observations.

First, concerning the evaluation of the boundary layer we have added a reference to the paper of LeMone et al (2013): *'The boundary-layer depth is a useful diagnostic to evaluate the representation of boundary-layer evolution in models as it results from the interplay of surface flux, turbulence and subsidence (LeMone et al, 2013).'* Concerning the PBL depth diagnostics as explained in the answer to P8 L30 we decided to keep the tke based diagnostic due to the better behaviour during the afternoon transition.

P16, bottom. Have you an idea what causes different types of morning PBL growth? Subsidence? Strength of inversion? And as you note, different criteria can give you different PBL depths. One you might mention is RH max, which will give you the top of the PBL in the absence of cumulus clouds, but will give you cloud base in the presence of cumulus clouds. (Curiously, we have found

that PBL turbulence statistics scale well with cloud base, but it can be argued that the true PBL depth is somewhere in the cumulus cloud layer.)

The rapid growth in the morning is due to presence of a residual layer that remained close to neutral as for instance for the 1 July (cf Lothon et al (2014); Blay-Carreras et al (2014)). The days with limited growth have smaller sensible heat fluxes and experience subsidence of warm air that made it very difficult for the CBL to grow. *We have now stated that 'The causes for different types of morning PBL growth is related to initial profiles, intensity of the sensible heat fluxes and of the subsidence as explained in Lothon et al (2014).'*

P17, L1. Isn't the top of the stable layer and the top of the inversion layer the same thing?

We change 'top of the inversion layer' to 'top of the residual layer'. The idea, here, is that the measurement during the afternoon can either detect the top of the stable boundary layer that is forming or the top of the residual layer corresponding to the trace of the convective boundary layer of that day.

P17, L7. Better prediction for AROME makes sense to me if you include shading, which could be a factor in decreasing surface buoyancy flux, especially near sunset. Do you?

I don't exactly understand what you meant by shading. Concerning shading by the orography, Orographic shading is now included in the operational AROME version, but it was not in the 2011 version used during BLLAST. But as the Pyrénées are mainly oriented East-West we do not expect a strong impact in the late afternoon as the sun set westward which will induce very small shading. Concerning shading by the vegetation, this is not included in the code. Concerning the shading by the clouds, this is included in the radiation code but the cloud fraction for each grid is the variable used in the radiation code, similarly as in ARPEGE. Both shading are complicated to take into account at 2.5km resolution.

P17 L16. "unique feature" rather than "specificity."

Done

Figure 8. Bigger font on right side. I can barely read the labels in my printed version – I'm working off my computer screen.

This figure has been redrawn and the labels have been enlarged. Sorry for that.

P17L30-31. Measured ON the evening … and IS reproduced.

Done

P 18. How do model and observed TKE compare if you count the TKE associated with resolved PBL eddies in AROME? And including some representation of the mass flux associated with the PBL scheme?

P18 L7, L14-15 While the difference in height might be a factor (at the lowest level), you would expect more parameterized TKE in ARPEGE compared to AROME because the PBL scheme in ARPEGE has to do almost all the transport (because the resolved eddies grow much more slowly for 10-km grid than for 2.5-km grid), see Ching et al. 2014, also LeMone et al. 2013). If it's an EDMF scheme, it should account for all the TKE. (Again one has to include somehow the mass flux in the TKE estimate).

P18, L19-20. Resolved PBL eddies grow during the day until saturation is reached. It could be that horizontally-averaged model TKE starts to go down as the resolved eddies grow. Though I am obviously skeptical that you will even achieve exact agreement, it is encouraging that the trends are similar.

There are no resolved PBL eddies in AROME. This might be due to the fact that the effective resolution is ~9 Dx (Ricard et al, 2013) which is 23.5 km (as also said to the answer of the general comment). The mass flux contribution is significant in the middle of the boundary layer and is now

accounted for in the analysis.

P19L5. "physical processes … are small ".. You mean terms in the TKE equation are small? If you don't want to write out the equation, you could write something like "Most of the terms in the TKE equation, -- buoyancy production, shear production, dissipation – are small." ? Or are all the terms small?
You are right. We modified the text with your suggestion: '*Most of the terms in the TKE equation- buoyancy production, shear production, dissipation, vertical transport- are small (Nilsson et al, 2016).*'

P19L15. "where the height of the reflectivity gradient decreases with time …"?
We have changed the sentence to ' *where the height of the reflectivity gradient decreases with time in the evening*'

P19L25-6. This makes sense, since dissipation and TKE are closely related.
We have added this comment in the text: '*… , which makes sense as tke and dissipation rate are closely related.*'

P20 paragraph 1. The earlier time at which sensible heat flux goes negative at the surface is consistent with large latent heat fluxes. This makes sense both from the point of view that more of the total energy is going into LH. But it also means that the buoyancy flux remains positive after the sensible heat flux goes negative. I would guess that the time when the buoyancy flux goes negative is also earlier for AROME, and this would be more directly related to turbulence generation than sensible heat flux. It would be good to see what a plot similar to Figure 10 for buoyancy flux looks like.
After your first quick review, we have modified the Figure 10 to show the time when the buoyancy flux becomes negative. Indeed, for most of the cases this delays the time of about 5-15 min; however the meaning of the figure is unchanged. Figure 10 of the revised manuscript is computed with the buoyancy flux as the buoyancy (and not the sensible heat flux) is the term that controls the intensity of turbulence (also see response to point 4 of your previous quick review showing both figures).

P20 paragraph 2.Because of the large latent heat fluxes, it might be useful to normalize thing in terms of buoyancy flux rather than sensible heat flux.
I am sorry but I did not understand what should be normalized by buoyancy flux. As explained above the time of the end of the transition, namely the time at which heat flux goes negative is now based on buoyancy flux instead of sensible heat flux.

P20L32-P21L1, suggest .. Models and observations produce lower sensible heat fluxes, higher temperatures, stronger winds, and weaker TKE than (what? For the other days?).
Following your advice we have changed the text to '*For instance, during the hot period, models and observations produce lower sensible heat fluxes, higher temperature, stronger winds, and weaker tke than during the other days.*'

P21 L22-3. From P7, L12-14, I thought that ARPEGE had the same PBL scheme as AROME. This sentence implies there is no "MF" in the ARPEGE PBL scheme. This could be clarified by listing the PBL physics schemes in a table and describing them more carefully. If there is no "MF" in the ARPEGE scheme, then the TKE should be pretty comparable to the total (no high-pass filtering) TKE. (Though I would expect some discrepancy since pure TKE schemes don't really represent what is going on in the CBL). Please clarify. As noted earlier, the parameterized TKE in an EDMF scheme should be smaller than measured, particularly for fine-mesh model runs (because of the

contribution of resolved eddies to the TKE).

This was not clear enough in the manuscript. ARPEGE and AROME do have the same *tke* scheme (Cuxart et al, 2000). However, in AROME, there is also a mass-flux scheme, based on the eddy-diffusivity mass-flux concept (Pergaud et al, 2009). In ARPEGE, there is only a mass-flux scheme active when shallow cumulus are present so this scheme is not active for clear days. There is already a table (table 2) that describes the different parameterizations; it was added after your first short review. As said previously there is no contribution of resolved eddies to the *tke* in AROME.

P21, end of 2nd paragraph. Estimation of some terms in the TKE budget might be simpler than the estimation of the TKE, at least in terms of direct comparison of model with observations, for reasons discussed earlier.

We agree with you. This is what we propose as a next step for this study. This is complicate to handle as all the runs have to be redone as the different TKE budget terms were not saved.

Supplementary Figure 2. Is this discussed?

This figure was discussed in the appendix but we decided to suppress the appendix.

P10, bottom. You refer to an Appendix here (which isn't part of the paper). Perhaps you should just refer to supplementary figure 2? Was there an appendix?

It seems that you may have had access to an earlier version of the text, possibly the one initially submitted before your earlier review. But after your quick review, we provided a new document. The appendix was added during this first step of revision but eventually we decided to suppress it.

References:

Cuxart J, Bougeault P, Redelsperger, JL.: A turbulence scheme allowing for mesoscale and large-eddy simulations. *Q. J. R. Meteorol. Soc.* 126 : 1-30, 2000

Giard D, Bazile E, 2000: Implementation of a new assimilation scheme for soil and surface variables in a global NWP model. Mon Wea Rev, 128, 997-1015

Lothon M,et al: The BLLAST field experiment: Boundary-Layer Late Afternoon and Sunset Turbulence, ACP, 2014

Pergaud J, Masson V, Malardel S, Couvreux F.: A parameterization of Dry thermals and shallow cumuli for mesoscale numerical weather prediction. *Boundary-Layer Meteorology*. **132**, 83-106. DOI 10.1007/s10546-009-9388-0, 2009

Responses to Earlier Review:
General comments
The paper is interesting and I think will be in a publishable form with some
modifications. I include here only some major thoughts (in no particular order).

1. The SH-vs-LH graphs having a slope close to -1 doesn't indicate a constant Bowen ratio- quite the opposite, it shows horizontal variation in Bowen ratio. Rather, it shows a constant available energy (i.e. LH+SH=constant). This is discussed for CASES-97 in LeMone et al 2003 (J Hydromet, choosing the averaging interval) and discussed as a function of soil moisture using both observations and a land-surface model in LeMone et al (2007). It is nice to see someone exploring this.

You are right. This was a mistake and has been changed in the text to ' the models reproduce the -1 slope related to an almost constant available energy (cf Supplementary Fig 1) in agreement with LeMone et al (2003).' We now quote the reference to LeMone et al 2003 that inspired us for drawing such graph.

2. When discussing horizontal heterogeneity, terrain plays a big role. It would be good for the authors to show maps of the terrain used in the three NWP models.

a. This is true, as the authors recognize, because of the presence of mountain-valley circulations of tall types. The different terrains will produce different circulations

b. This is also true for horizontal variability. Although one gets downslope drainage winds event with gentle terrain, more extreme terrain probably has more cold-air pooling. So, there might be less horizontal variability smoothed terrain.

In Table 3, the altitude of each point is indicated, which provides information on the resolved orography in the models for the points used for the intercomparison. According to your comment, we have included below the map of the resolved orography on a small domain around the observation sites for the ECMWF model as well as the AROME model, those two models having the coarsest and the finest horizontal resolution. Indeed, the orography is better resolved in the finer resolution. The impact of the mountain-valley circulation is the subject of an ongoing study.

[Figure]

Figure a: orography of the models (in m) in (left) ECMWF operational model and (right) AROME operational model.

3. When discussing TKE, the measurements will inevitably include the impact of large eddies (horizontal wavelength between 1.5 and 3 times the depth of the PBL roughly). These are likely partially resolved in AROME and they tend to grow in models under convective conditions, faster with smaller grid spacing. Comparison to TKE in the other two models is in some sense more realistic from this point of view, since neraly all TKE will be parameterized. For fine grid spacing (< ~4km), the interaction between this resolved eddies and PBL schemes can exaggerate local variations in TKE (see Ching et al. MWR, 2014 and references therein).

Other issues :

a. TKE is mostly horizontal near the surface, especially for eddies extending through the PBL (for which w is very small ; from mass-continuity equation).

b. Large eddies travel roughly at the mean speed of the wind through their depth (i.e. the boundary layer). Thus if one filters according to scale, the scale should be defined not be the wind at the level of the measurement, but by the mean PBL wind.

c. A philosophical point (discussed in Ching et al) is the in the 'gray zone' or 'terra incognita' the PBL scheme should account for all the TKE in the PBL, which for fine-grid models mean several grid points horizontally, and there should be no large PBL eddies (convective rolls or cells). (This is the purists' view ; the semi-resolved eddies have been useful in storm initiation or propagation – because large eddies, especially rolls, have been shown to play a role in storm propagation and evolution). One way to look at this is by considering the buoyancy-flux profile. It should be continuous from the surface (where its value is determined by a land-surface model) up through the PBL. If one does time- or space- filtering that is too fine, the fluxes above the surface are too small. I gather from the discussion that the authors were wrestling with this.

Thanks for the reference. The 'gray zone' is indeed an issue for a model at kilometric scale as shown in Honnert et al (2011). However, here, the runs performed with the finest resolution (AROME

model) have a horizontal grid of 2.5km and an effective resolution of about 9Dx(~22km, cf Ricard et al, 2013). Therefore this is not an issue for this model and the turbulence is still fully subgrid. We have checked the buoyancy-flux profile as you proposed and also checked in horizontal cross section that no spurious numerical convective rolls or cells occurred. Note that, in AROME, the boundary-layer turbulence is handled by a EDMF scheme with the ED component being represented with a prognostic turbulent kinetic energy scheme and the MF component being handled by a mass-flux scheme which introduces a non-local contribution as advised by Ching et al. Not all the turbulence is handled by the turbulent kinetic energy scheme, part of the turbulence (the non-local thermals) is handled by the mass-flux scheme, this is what we discuss in the paper.

4. The authors should look at the paper by Lindsey Bennett et al. (MWR, 2010) regarding estimates from different instrumentation, in addition to the LeMone et al, and Grimsdell and Angevine work cited.
We have added a reference to this paper in the manuscript relative to the various estimates of boundary-layer depth in page 6 of the new manuscript: *'The comparison of different boundary-layer depths derived from various instruments has been illustrated in Bennett et al (2010).'* and in page 7 of the new manuscript '*The decrease of the boundary-layer depth in the afternoon transition is a delicate process and in practice, its estimation depends on the criteria used to derive the boundary-layer depth as already shown by Angevine and Grimsdell (2002) and Bennett et al (2010).'*

5. Figure 10 : should look at the time when the virtual temperature flux becomes negative in the afternoon ; or, similarly, how this time relates to latent heat flux (and hence vegetation type and soil moisture). If I recall correctly, the sensible heat flux went negative earlier where there was large latent heat flux based on CASES-97 data. Which meant more variability in this time both spatially and from day to day for sensible heat flux than for virtual temperature flux.
Thanks for this comment. We redrew Figure 10 with the virtual temperature flux. Below you can find both figures, in the left hand side, the time when the temperature flux becomes negative and in the right hand side, the time when the virtual temperature flux becomes negative. Indeed, the time when the virtual temperature flux goes negative is later than the time when the temperature flux becomes negative and the scatter is reduced. We have now included in Figure 10 the one computed with the virtual temperature flux.

[Figure]

General editorial comments (more detail later) :
1. The figures are impossible to study in printed form. I am reviewing the paper with the figures enlarged on the screen. The labels on the right hand side need to be larger, and 'range' might be a better label than 'variability'. Also it would be helpful to the reader to label the 'hot' days referred to in the text. And have labels on all the figures to make it easier for the reader.
We have modified the figures accordingly. The labels on the right hand side of Figures 2, 3, 4 are larger and we have replaced 'variability' by 'range'. We have also added labels for the days in all the plots and a label indicating the hot days.

2. It would help in the profile figure (6) to figures to have grid points on the curves for each model – for one curve for each model. Also, might consider plotting the average profile for each model and time. And finally might consider offsetting the soundings by adding a few degrees for each time interval. The last might not be practical (You could stretch the horizontal axis and only have one altitude label).

We have plotted the average profile for each model and time which is shown with dashed lines and a slightly different color than the profiles for each point. We have also added a 2K and a 2g/kg offset for each time interval in order to better distinguish the different hours.

3. It would be useful to have a table describing the properties of each model (horizontal and vertical grid spacing, PBL scheme, etc.) as well as model-run length and initiation time and data used to initialize the model. Also how the land-surface properties were initialized (often there is a long-spinup).

We have added a Table (now Table 2) describing the properties of each model.

References:

Honnert T, V Masson, F Couvreux, 2011: A diagnostic for evaluating the representation of turbulence in atmospheric models at kilometric scale. J Atmos Sci, 68, 3112-3131

Ricard, D., Lac, C., Riette, S., Legrand, R. and Mary, A. (2013), Kinetic energy spectra characteristics of two convection-permitting limited-area models AROME and Meso-NH. Q.J.R. Meteorol. Soc., 139: 1327–1341. doi: 10.1002/qj.2025

---

## Author Comment (AC2) · 20 Jun 2016

**Answer to the reviewer 2 about the manuscript entitled : 'Boundary-layer turbulent processes and mesoscale variability represented by Numerical Weather Prediction models during the BLLAST campaign' by F. Couvreux et al.:**

First, we wish to thank the reviewer for her/his review. Below is our response (in blue) to the comments on a point-by-point basis. References to how we plan to modify the text is indicated in italic.

General comments:

After careful reading, my general impression is that the manuscript contains relevant and sound scientific findings as result of a massive analysis work, and deserves publication. It is a pity, though, that the exposure of such a wealth of results is rather poor. The reading is hard and fragmented, with too many inaccuracies and repetitions. The style needs improvement before the paper can be accepted for publication. To my opinion, the figures are simply not to the ACP standard and require a complete rethinking, not only for publication but even for review. I have struggled to get useful information out of the figures in their current format. I leave the editor the decision if they can be accepted as they are. The structure and 'paragraphing' used for the discussion of the results, on the other hand, seems appropriate.

We have tried to improve the layout of the paper by improving the quality of the figures and trying to suppress repetitions and inaccuracies. Figure 1 has been enlarged to be more visible. A new Figure 2 has been added to present the orography in the real world and in the different models. In particular, old figures 2, 3 and 4 have been simplified and now only show the mean curves as you proposed (cf answer to your specific comment below). A figure presenting the overestimation of the sensible heat fluxes in ARPEGE has also been added and is presented with error bars to show the standard deviation in the observations. The colors of Figure 7 have been modified. We suppressed the appendix for clarity. In addition, a native English speaker has checked out the entire manuscript. We hope that now the paper reads more easily.

Specific comments:

First two lines of page 3. I don't understand the meaning of the sentence. Can you please clarify? Those two lines have been modified to '*Several recent studies also assessed the behaviour of single-column models to represent the entire diurnal cycle by comparison to LES.*'

Second line of page 3. '. . .single-column runs ARE often used as a simplified configuration OF a full 3D simulation ...'. Also, define 'single-column' models. Done, we now define single-column model as *one single column of the atmosphere that integrates the same suite of parameterizations as a full 3D simulation*.

Line 10 page 3. '. . .are quite rare compared to. . .' Done

Define tke the first time you introduce the turbulent kinetic energy and (same for IOP). OK This is now done on page 2 line 7 for the turbulent kinetic energy. IOP and tke are introduced twice, once first in the abstract and a second time in the main text.

Remove 'days' after IOP. Done throughout the text

Page 4, line 7.' . . . all surface stations measuring turbulence. . .' Done

The first two lines of section 2.3 can be removed, or at least, rephrased. 'Due to the coarse grid spacing. . .' Following your advice, we have rephrased the sentence as : « *Due to the coarse grid spacing of each model, real surface heterogeneities, topography and local circulation are not expected to be reproduced by models.*"

Page 6, line 15. '. . .the tke is below . . .. In the observations. . .' Done

Page 6, line 18. '. . .usually provides an estimate. . ., based on the vertical gradient of the relative humidity'  Done

Page 7, line 5. '. . .at a given hour h correspond. . .'  Done

The paragraph at the beginning of section 3 should be moved to the methodology section Page 11.
As you proposed, we moved the paragraph at the beginning of the section 3 to the end of the methodology section (in the part 2.3).
line 12. '. . .variables indicating different. . .'  Done

Page 11, line 26. '. . . the boundary layer depth estimated by the model with the boundary layer depth estimated by the observations'. Done

Page 11, line 29. Please provide reference  We have added in the methodology section some words on the comparison of the boundary-layer height diagnosed from the tke profiles versus boundary-layer height diagnosed from thermodynamical profiles.

Page 11, line 30. '. . .the temporal variability in terms of maximum boundary layer depth from on a day to the other. . .' is not clear. Do you mean the variability diurnal cycle?
We wanted to comment in terms of variability from one day to the other of the maximum boundary-layer depth of the day, so not the diurnal cycle. This has been changed to '*Both AROME and ARPEGE are able to reproduce days with higher boundary layers compared to days with shallower boundary layers with for instance a shallower boundary layer during the hot days and, the highest on 30 June, 1 July and 2 July if we discard the 14 and 15 June.*'

Page 11, line 34. '. . .the physics of the models respondS . . .'  Done

The end of page 8 is a left-over of some copy-paste?
We are sorry that this happens. In fact, at the last minute, we had to use another template provided by the journal. In fact this was note a left-over of some copy-paste but a foot note. However, due to no changes in the police, neither in its side, it really looks like a copy-paste. We decided to suppress the footnote and included the information into parenthesis in the main text.

The first sentence of section 3.3 is unnecessary (already said a few times) We suppressed this sentence.
In the Appendix the last words sounds strange..' A=3 would be a value too large'.
Eventually, we decided to suppress the Appendix and therefore the supplementary 2.

Table 2. The roughness length is measured in meters
You are right. We have added the unit in this table.

Figures 2. I would suggest to keep only the mean curves and/or to replace the time series with box and whiskers, four for each IOP (obs plus three models) or three is you prefer to plot the bias (obs – mods). Add the legend to all figures if possible, to help the readers.
As you propose, we decided to only keep the figures showing the mean curves.  We also added a figure to show the over-estimation of the sensible heat flux for ARPEGE point (now Figure 3). In this figure, we have use bars to indicate the horizontal standard deviation of observations. Eventually, now a complete caption is present for each figure.

Figure 6. The choice of colours is unfortunate. Why not blue, red and green for example? The graph is anyway difficult to interpret, please try to make clearer (in the caption please use 'becomes' in place of 'goes').

We have changed the colours. The colours are chosen here to reflect the daytime to nighttime evolution with red for 1200 profile, orange for 1400, dark-green for 1600, purple for 1800 and grey for 2000. We also changed 'goes' to 'becomes'.